# SUFFICIENT CONDITIONS FOR ROBUSTNESS TO ADVERSARIAL EXAMPLES: A THEORETICAL AND EMPIRICAL STUDY WITH BAYESIAN NEURAL NETWORKS

## ABSTRACT

We prove, under two sufficient conditions, that idealised models can have no adversarial examples. We discuss which idealised models satisfy our conditions, and show that idealised Bayesian neural networks (BNNs) satisfy these. We continue by studying near-idealised BNNs using HMC inference, demonstrating the theoretical ideas in practice. We experiment with HMC on synthetic data derived from MNIST for which we know the ground-truth image density, showing that near-perfect epistemic uncertainty correlates to density under image manifold, and that adversarial images lie off the manifold in our setting. This suggests why MC dropout, which can be seen as performing approximate inference, has been observed to be an effective defence against adversarial examples in practice; We highlight failure-cases of non-idealised BNNs relying on dropout, suggesting a new attack for dropout models and a new defence as well. Lastly, we demonstrate the defence on a cats-vs-dogs image classification task with a VGG13 variant.

## 1 INTRODUCTION

Adversarial examples, inputs to machine learning models that an adversary designs to manipulate model output, pose a major concern in machine learning applications. Many hypotheses have been suggested in the literature trying to explain the existence of adversarial examples. For example, Tanay & Griffin (2016) hypothesise that these examples lie near the decision boundary, while Nguyen et al. (2015) hypothesise that these examples lie in low density regions of the input space. However, adversarial examples can lie far from the decision boundary (e.g. "garbage" images (Nguyen et al., 2015)), and using a simple spheres dataset it was shown that adversarial examples can exist in high density regions as well (Gilmer et al., 2018). In parallel work following Nguyen et al. (2015)'s low-density hypothesis, Li (2018) empirically modelled input image density on MNIST and successfully detected adversarial examples by thresholding low input density. This puzzling observation, seemingly inconsistent with the spheres experiment in (Gilmer et al., 2018), suggests that perhaps additional conditions beyond the ability to detect low input density have led to the observed robustness by Li (2018).

Suggesting two sufficient conditions, here we prove that an idealised model (in a sense defined below) cannot have adversarial examples, neither in low density nor in high density regions of the input space. We concentrate on adversarial examples in discriminative classification models, models which are used in practical applications. To formalise our treatment, and to gain intuition into the results, we use tools such as discriminative Bayesian neural network (BNN) classifiers (MacKay, 1992; Neal, 1995) together with their connections to modern techniques in deep learning such as stochastic regularisation techniques (Gal, 2016). This pragmatic Bayesian perspective allows us to shed some new light on the phenomenon of adversarial examples. We further discuss which models other than BNNs abide by our conditions. Our hypothesis suggests *why* MC dropout-based techniques are sensible for adversarial examples identification, and why these have been observed to be consistently effective against a variety of attacks (Li & Gal, 2017; Feinman et al., 2017; Rawat et al., 2017; Carlini & Wagner, 2017).

We support our hypothesis mathematically and experimentally using HMC and dropout inference. We construct a synthetic dataset derived from MNIST for which we can calculate ground truth input densities, and use this dataset to demonstrate that model uncertainty correlates to input density, and that under our conditions this density is low for adversarial examples. Using our new-found insights we develop a new attack for MC dropout-based models which does not require gradient information, by looking for "holes" in the epistemic uncertainty estimation, i.e. imperfections in the uncertainty

approximation, and suggest a mitigation technique as well. We give illustrative examples using MNIST (LeCun & Cortes, 1998), and experiment with real-world cats-vs-dogs image classification tasks (Elson et al., 2007) using a VGG13 variant (Simonyan & Zisserman, 2015).

## 2 RELATED LITERATURE

There has been much discussion in the literature about the nature of "adversarial examples". Introduced in Szegedy et al. (2013) using gradient crafting techniques for image inputs, these were initially hypothesised to be similar to the rational numbers, a dense set within the set of all images[1]. Szegedy et al. (2013)'s gradient based crafting method performed a *targeted* attack, where an input image is perturbed with a small perturbation to classify differently to the original image class. Follow-up research by Goodfellow et al. (2014) introduced *non-targeted* attacks, where a given input image is perturbed to an arbitrary wrong class by following the gradient away from the image label. This crafting technique also gave rise to a new type of adversarial examples, "garbage" images, which look nothing like the original training examples yet classify with high output probability. Goodfellow et al. (2014) showed that the deep neural networks' (NNs) non-linearity property is not the cause of vulnerability to adversarial examples, by demonstrating the existence of adversarial examples in linear models as well. They hypothesised that NNs are very linear by design and that in high-dimension spaces this is sufficient to cause adversarial examples. Later work studied the linearity hypothesis further by constructing linear classifiers which do not suffer from the phenomenon (Tanay & Griffin, 2016). Instead, Tanay & Griffin (2016) argued that adversarial examples exist when the classification boundary lies close to the manifold of sampled data.

After the introduction of adversarial examples by Szegedy et al. (2013), Nguyen et al. (2015) developed crafting techniques which do not rely on gradients but rather use genetic algorithms to generate "garbage" adversarial examples. Nguyen et al. (2015) further hypothesised that such adversarial examples have low probability under the data distribution, and that joint density models $p(\mathbf{x}, y)$ will be more 'robust' because the low marginal probability $p(\mathbf{x})$ would be indicative of an example being adversarial. Nguyen et al. (2015) argued that this mitigation is not practical though since current generative models do not scale well to complex high-dimensional data distributions such as ImageNet. Li (2018) recently extended these ideas to non-garbage adversarial examples as well, and lent support to the hypothesis by showing on MNIST that a deep naive Bayes classifier (a generative model) is able to detect targeted adversarial examples by thresholding low input density. Parallel work to Li (2018) has also looked at the hypothesis of adversarial examples having to exist in low input density regions, but proposed that adversarial examples *can* exist in high density regions as well. More specifically, Gilmer et al. (2018) construct a simple dataset composed of a uniform distribution over two concentric spheres in high dimensions, with a deterministic feed-forward NN trained on 50M random samples from the two spheres. They propose an attack named "manifold attack" which constrains the perturbed adversarial examples to lie on one of the two concentric spheres, i.e. in a region of high density, and demonstrate that the attack successfully finds adversarial examples with a model trained on the spheres dataset. This demonstration that there *could exist* adversarial examples on the data manifold and in high input density regions falsifies the hypothesis that adversarial examples must exist in low density regions of the input space, and is seemingly contradictory to the evidence presented in Li (2018). We will resolve this inconsistency below.

A parallel line to the above research has tried to construct bounds on the minimum magnitude of the perturbation required for an image to become adversarial. Fawzi et al. (2018) for example quantify "robustness" using an introduced metric of expected perturbation magnitude and derive an upper bound on a model's robustness. Fawzi et al. (2018)'s derivation relies on some strong assumptions, for example assuming that it is feasible to compute the distance between an input $\mathbf{x}$ and the set $\{\mathbf{x} : f(\mathbf{x}) > 0\}$ for some classifier $f(\mathbf{x})$. Papernot et al. (2016) further give a definition of a robust model, extending the definitions of Fawzi et al. (2018) to targeted attacks, and propose a model to satisfy this definition. In more recent work, Peck et al. (2017) extend on both these ideas (Fawzi et al., 2018; Papernot et al., 2016), and propose a lower bound on the robustness to perturbations necessary to change the classification of a neural network. Peck et al. (2017) also make strong assumptions in their premise, assuming the existence of an oracle $f^*(\mathbf{x})$ able to assign a "correct" label for each input $\mathbf{x} \in \mathbb{R}^D$. This assumption is rather problematic since it implies that any input has

---

[1]This was refuted in (Goodfellow et al., 2014); Below we will see another simple theoretical argument refuting this hypothesis.

a "correct" class, including completely blank images which have no objects in them. Lastly, Hein & Andriushchenko (2017), working in parallel to (Peck et al., 2017), use alternative assumptions and instead offer a bound relying on local Lipschitz continuity.

Following the perturbation bounds literature, in this work we will use similar but simpler tools, relying on the continuity of the classifier alone. Contrary to the generative modelling perspective, we will concentrate on *discriminative* Bayesian models which are much easier to scale to high-dimensional data (Kendall & Gal, 2017). Such models capture information about the density of the training set as we will see below. We will define our idealised models under some strong assumptions (as expected from an *idealised* model), in a similar fashion to previous research concerned with provable guarantees. However below we will also give empirical support demonstrating the ideas we develop with practical tools. The class of models which satisfy our conditions postulated below includes models other than BNNs, such as RBF networks and nearest neighbour in feature space. Even so, we will formalise our arguments in 'BNN terminology' to keep precise and rigorous language. After laying out our ideas, below we will discuss which other models our results extend to as well.

## 3 BACKGROUND

A deep neural network for classification is a function $f : \mathbb{R}^D \mapsto \mathcal{Y}$ from an input space $\mathbb{R}^D$ (e.g. images) to a set of labels (e.g. $\{0, 1\}$). The network $f$ is parametrised by a set of weights and biases $\omega = \{\mathbf{W}_l, \mathbf{b}_l\}_{l=1}^L$, which are generally chosen to minimize some empirical risk $E : \mathcal{Y} \times \mathcal{Y} \mapsto \mathbb{R}$ on the model outputs and the target outputs over some dataset $\mathbf{X} = \{\mathbf{x}_i\}_{i=1}^N, \mathbf{Y} = \{y_i\}_{i=1}^N$ with $\mathbf{x}_i \in \mathbb{R}^D$ and $y_i \in \mathcal{Y}$. Rather than thinking of the weights as fixed parameters to be optimized over, the Bayesian approach is to treat them as random variables, and so we place a prior distribution $p(\omega)$ over the weights of the network. If we also have a likelihood function $p(y \mid \mathbf{x}, \omega)$ that gives the probability of $y \in \mathcal{Y}$ given a set of parameter values and an input to the network, then we can conduct *Bayesian inference* given a dataset by marginalising (integrating out) the parameters. Such models are known as *Bayesian neural networks* (MacKay, 1992; Neal, 1995). The conditional probability of the model parameters $\omega$ given a training set $\mathbf{X}, \mathbf{Y}$ is known as the posterior distribution. Ideally we would integrate out our uncertainty by taking the expectation of the predictions over the posterior, rather than using a point estimate of the parameters (e.g. MAP, the maximiser of the posterior). For deep Bayesian neural networks this marginalisation cannot be done analytically. Several approximate inference techniques exist, and here we will concentrate on two of them. Hamiltonian Monte Carlo (HMC, Neal, 1995) is considered to be the 'gold-standard' in inference, but does not scale well to large amounts of data. It has been demonstrated to give state-of-the-art results on many small-scale tasks involving uncertainty estimation in non-tractable models (Neal, 1995). A more pragmatic alternative is approximate variational inference, e.g. with dropout approximating distributions (Gal, 2016). This technique is known to scale to large models, preserving model accuracy, while giving useful uncertainty estimates for various down-stream tasks (Kendall & Gal, 2017). However, dropout approximate inference is known to give worse calibrated approximating distributions, a fact we highlight below as well.

Bayesian neural networks are tightly connected to Gaussian processes (Rasmussen & Williams, 2006), and in fact the latter Gaussian processes can be seen as the infinite limit of single hidden layer Bayesian neural networks with Gaussian priors over their weights (Neal, 1995). Both can quantify "epistemic uncertainty": uncertainty due to our lack of *knowledge*. In terms of machine learning, this corresponds to a situation where our model output is poorly determined due to lack of data near the input we are attempting to predict an output for. This is distinguished from "aleatoric uncertainty" (which we will refer to below as *ambiguity*) which is due to genuine stochasticity in the data (Kendall & Gal, 2017): This corresponds to *noisy* data, for example digit images that can be interpreted as either 1 or 7; no matter how much data the model has seen, if there is inherent noise in the labels then the best prediction possible may be a high entropy one (for example, if we train a model to predict fair coin flips, the best prediction is the max-entropy distribution $P(\text{heads}) = P(\text{tails})$).

An attractive measure of uncertainty able to distinguish epistemic from aleatoric examples is the information gain between the model parameters and the data. Recall that the mutual information (MI) between two random variables (r.v.s) $X$ and $Y$ is given by

$$\mathcal{I}(X, Y) = \mathcal{H}[X] - \mathbb{E}_{P(y)}[\mathcal{H}[X \mid Y]] = \mathcal{H}[Y] - \mathbb{E}_{P(X)}[\mathcal{H}[Y \mid X]]$$

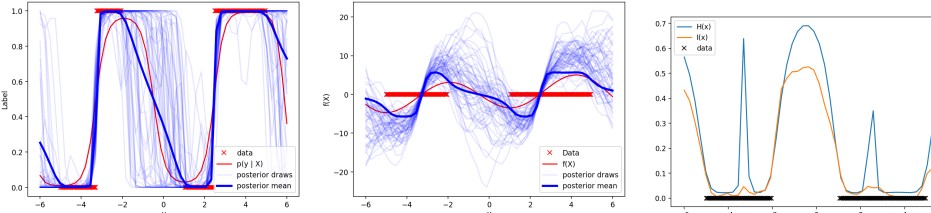

Figure 1: Illustration of function realisations in softmax space (left), in logit space (pre-softmax, middle), as well as epistemic (orange, right) and aleatoric uncertainty (blue, right). Note the high epistemic uncertainty ($\mathcal{I}$) in regions of the input space where many function explanations exist, and how predictive probability mean (dark blue, left panel) is close to uniform in these areas. Also note aleatoric uncertainty $\mathcal{H}$ spiking in regions of ambiguity (transition from class 0 to class 1, depicted in the left panel).

with $\mathcal{H}[X]$ the entropy of r.v. $X$. In terms of machine learning, the amount of information we would gain about the model parameters r.v. $\omega$ if we were to receive a label realisation for the r.v. $y$ for a new input $\mathbf{x}$, given the dataset $\mathcal{D}$, is then given by the difference between the predictive entropy $\mathcal{H}[y \mid \mathbf{x}, \mathcal{D}]$ and the expected entropy $\mathbb{E}_{p(\omega|\mathcal{D})}[\mathcal{H}[y \mid \mathbf{x}, \omega]]$:

$$\mathcal{I}(\omega, y \mid \mathcal{D}, \mathbf{x}) = \mathcal{H}[y \mid \mathbf{x}, \mathcal{D}] - \mathbb{E}_{p(\omega|\mathcal{D})}[\mathcal{H}[y \mid \mathbf{x}, \omega]]. \tag{1}$$

Being uncertain about an input point $\mathbf{x}$ implies that if we knew the label at that point we would gain information. Conversely, if the function output at an input is already well determined, then we would gain little information from obtaining the label. Thus, the MI is a measurement of the model's *epistemic* uncertainty (in contrast to the predictive entropy which is high when *either* the epistemic uncertainty is high *or* when there is ambiguity, e.g. refer to the example of a fair coin toss). Note that the MI is always bounded between 0 and the predictive entropy.

To gain intuition into the different types of uncertainty in BNNs we shall look at BNN realisations in function space with a toy dataset. Our BNN defines a distribution over NN parameters, which induces a distribution over functions from the input space to the output space. Drawing multiple function realisations we see (Fig. 1) that *all* functions map the training set inputs to the outputs, but each function takes a different, rather arbitrary, value on points outside the train set. Assessing the discrepancy of these functions on a given input allows us to identify if the tested point is near the training data or not. In classification, having high enough discrepancy between the pre-softmax functions' values for a fixed input leads to lower output probability when averaged over the post-softmax values. Thus any input far enough from the training set will have low output probability.

## 4 THEORETICAL JUSTIFICATION

We now show that *idealised* discriminative Bayesian neural networks, capturing perfect epistemic uncertainty and data invariances, cannot have adversarial examples. In §A we give some informal intuition into our proof, while in §B we look at the proof critically (these sections appear in the appendix due to space constraints). Here we follow (Nguyen et al., 2015; Papernot et al., 2016) where an adversarial example is defined as follows.

**Definition 1.** *An adversarial example is a model input which either*

1. *lies far from the training data but is still classified with high output probability (e.g. 'garbage' images), or*

2. *an example which is formed of an input $\mathbf{x}$ which classifies with high output probability, and a small perturbation $\eta$, s.t. a prediction on $\mathbf{x} + \eta$ is also made with high output probability, and the predicted class on $\mathbf{x}$ differs from the predicted class on $\mathbf{x} + \eta$. The perturbation $\eta$ can be either perceptible or imperceptible.*

Note that other types of adversarial examples exist in the literature (Yang et al., 2017; Grosse et al., 2017; Kreuk et al., 2018).

We start by setting our premise. We will develop our proof for a binary classification setting with continuous models (i.e. the model is *discriminative* and its output is a single probability $p$ between 0 and 1, continuous with the input $\mathbf{x}$), with a finite training set $\mathbf{X} \in \mathbb{R}^{N \times D}, \mathbf{Y} \in \{0, 1\}^N$ sampled from some data distribution. Our first assumption is that the training data has no ambiguity:

**Assumption 1.** *There exist no $\mathbf{x} \in \mathbf{X}$ which is labelled with both class 0 and class 1.*

This requirement of lack of ambiguity will be clarified below. We define an $\epsilon$ threshold for a prediction to be said to have been made 'with high output probability': $p > 1 - \epsilon$ is defined as predicting class 1 with high output probability, and respectively $p < \epsilon$ is said to predict class 0 with high output probability (e.g. $\epsilon = 0.1$ is a nice choice).

Our first definition is the set containing all transformations $\mathcal{T}$ that our data is invariant under, e.g. $\mathcal{T}$ might be a set containing translations and local deformations for image data:

**Definition 2.** *Let $p(\mathbf{x}, y)$ be the data distribution $\mathbf{X}, \mathbf{Y}$ were i.i.d. sampled from. Define $\mathcal{T}$ to be the set of all transformations $T$ s.t. $p(y|\mathbf{x}) = p(y|T(\mathbf{x}))$ for all $\mathbf{x} \in \mathbf{X}, y \in \mathcal{Y}$.*

Note that $\mathcal{T}$ cannot introduce ambiguity into our training set. For brevity *in the proof alone* (i.e. not for actual model training) we overload $\mathbf{X}$ and use it to denote the augmented training set $\{T(\mathbf{x}) : \mathbf{x} \in \mathbf{X}, T \in \mathcal{T}\}$, i.e. we augment $\mathbf{X}$ with all the possible transformations on it (note that $\mathbf{X}$ may now be infinite); We further augment and overload $\mathbf{Y}$ correspondingly so each $T(\mathbf{x}) \in \mathbf{X}$ is matched with the label $y$ corresponding to $\mathbf{x}$. Note that to guarantee *full* coverage (i.e. all input points with high probability for some $y$ under the data distribution must have high output probability under the model) one would demand $\mathcal{X}/\mathcal{T} \subseteq \mathbf{X}$, i.e. every point in the input space must belong to some trajectory generated by some point from the train set, or equivalently, all equivalence classes defined by $\mathcal{T}$ must be represented in the train set. We next formalise what we mean by 'idealised NN':

**Definition 3.** *We define an 'idealised NN' to be a NN which outputs probability 1 for each training set point $\mathbf{x} \in \mathbf{X}$ with label 1, and outputs probability 0 on training set points $\mathbf{x}$ with label 0. We further define a 'Bayesian idealised NN' to be a Bayesian model average of idealised NNs (i.e. we place a distribution over idealised NNs' weights).*

Note that this definition implies that the NN architecture is invariant to $\mathcal{T}$, **our first condition** for a model to be robust to adversarial examples. Model output (the predictive probability) for a Bayesian idealised NN is given by Bayesian model averaging: $p(y|\mathbf{x}, \mathbf{X}, \mathbf{Y}) = \int p(y|\mathbf{x}, \omega)p(\omega|\mathbf{X}, \mathbf{Y})\mathrm{d}\omega$, which we write as $\mathbf{p}_{\text{BNN}}(y|\mathbf{x})$ for brevity. Note that a Bayesian idealised NN must have predictive probabilities taking one of the two values in $\{0, 1\}$ on the training set.

Following Neal (1995) we know that infinitely wide (single hidden layer) BNNs converge to Gaussian processes (GPs, Rasmussen & Williams, 2006). In more recent results, Matthews et al. (2017) showed that even finite width BNNs with more than a single hidden layer share many properties with GPs. Of particular interest to us is the GP's epistemic uncertainty property (uncertainty which can increase 'far' from the training data, where far is defined using the GP's lengthscale parameter)[2]. We next formalise what we mean by 'epistemic uncertainty'.

**Definition 4.** *We define 'epistemic uncertainty' to be the mutual information $\mathcal{I}(\omega, y|\mathcal{D}, \mathbf{x})$ between the model parameters r.v. $\omega$ and the model output r.v. $y$.*

Denoting the model output probability $\mathbf{p}_{\text{BNN}}(y|\mathbf{x})$ by $p$, we abuse notation slightly and write $\mathcal{I}(\omega; p), \mathcal{H}(p)$ instead of $\mathcal{I}(\omega, y|\mathcal{D}, \mathbf{x}), \mathcal{H}(y|\mathbf{x})$ for our Bernoulli r.v. $y$ with mean $p$. Note that the mutual information satisfies $\mathcal{H}(p) \geq \mathcal{I}(\omega; p) \geq 0$. Since we assumed there exists no ambiguous $\mathbf{x}$ in the dataset $\mathbf{X}$, we have $\mathcal{H}(p) = \mathcal{I}(\omega; p)$ for all $\mathbf{x} \in \mathbf{X}$.

Next we introduce a supporting lemma which we will use in our definition of an 'idealised BNN':

**Lemma 1.** *Let $\mathbf{p}_{BNN}(y|\mathbf{x})$ be the model output of some Bayesian idealised NN on input $\mathbf{x} \in \mathbb{R}^D$ with training set $\mathbf{X}, \mathbf{Y}$. There exists $\delta_{BNN}^{\mathbf{x}}$ for each $\mathbf{x} \in \mathbf{X}$ such that the model predicts with high output probability on all $\mathbf{x}'$ in the delta-ball[3] $B(\mathbf{x}, \delta_{BNN}^{\mathbf{x}})$.*

*Proof.* Let $\mathbf{x} \in \mathbf{X}$ be a training point. By Bayesian idealised NN definition, $\mathbf{p}_{\text{BNN}}(y|\mathbf{x})$ takes a value from $\{0, 1\}$. W.l.o.g. assume $\mathbf{p}_{\text{BNN}}(y|\mathbf{x}) = 1$. By continuity of the BNN's output $\mathbf{p}_{\text{BNN}}(y|\mathbf{x})$ there exists a $\delta_{\text{BNN}}^{\mathbf{x}}$ s.t. all $\mathbf{x}'$ in the delta ball $B(\mathbf{x}, \delta_{\text{BNN}}^{\mathbf{x}})$ have model output probability larger than $1 - \epsilon$. Similarly for $\mathbf{p}_{\text{BNN}}(y|\mathbf{x}) = 0$, there exists a $\delta_{\text{BNN}}^{\mathbf{x}}$ s.t. all $\mathbf{x}'$ in the delta ball $B(\mathbf{x}, \delta_{\text{BNN}}^{\mathbf{x}})$ have model output probability smaller than $\epsilon$. I.e. the model output probability is as that of $\mathbf{p}_{\text{BNN}}(y|\mathbf{x}) \in \{0, 1\}$ up to an $\epsilon$, and the model predicts with high output probability within the delta-ball. $\square$

---

[2]Note that this property depends on the GP's kernel; we discuss this in the next section.
[3]A delta ball around $\mathbf{x}$ is defined as $B(\mathbf{x}, \delta) = \{\mathbf{x}' \in \mathbb{R}^D : ||\mathbf{x}' - \mathbf{x}||_2 < \delta\}$.

Finally, we define an 'idealised BNN' to be a Bayesian idealised NN which has a 'GP like' distribution over the function space (where the GP's kernel should account for the invariances $\mathcal{T}$ which are built into the BNN model architecture, see for example (van der Wilk et al., 2017)), and which increases its uncertainty 'fast enough'. Or more formally:

**Definition 5.** *We define an idealised BNN to be a Bayesian idealised NN with epistemic uncertainty higher than $\mathcal{H}(\epsilon)$ outside $\mathcal{D}'$, the union of $\delta^{\mathbf{x}}_{BNN}$-balls surrounding the training set $\mathbf{x}$ points.*

**This is our second condition** which must be satisfied for a model to be robust to adversarial examples. We now have the tools required to state our main result:

**Theorem 1.** *Under the assumptions and definitions above, an idealised Bayesian neural network cannot have adversarial examples.*

*Proof.* Let $\mathbf{x} \in \mathbf{X}$. By lemma 1, every perturbation $\mathbf{x} + \eta$ that is under the delta ball $B(\mathbf{x}, \delta^{\mathbf{x}}_{BNN})$ does not change class prediction. Further, by the idealised BNN definition and epistemic uncertainty definition, we have that for all $\mathbf{x}'$ outside $\mathcal{D}'$, with the model output probability on $\mathbf{x}'$ denoted as $\mathbf{p}_{BNN}(y|\mathbf{x}')$, the entropy satisfies $\mathcal{H}(\mathbf{p}_{BNN}(y|\mathbf{x}')) \geq \mathcal{I}(\omega; \mathbf{p}_{BNN}(y|\mathbf{x}')) > \mathcal{H}(\epsilon)$. By symmetry, entropy of $\mathbf{p}_{BNN}(y|\mathbf{x}')$ being larger than the entropy of $\epsilon$ means that $\epsilon \leq \mathbf{p}_{BNN}(y|\mathbf{x}') \leq 1 - \epsilon$, i.e. the prediction is with low output probability for both class 0 and class 1.

We have that every $\mathbf{x} \in \mathbb{R}^D$ has either 1) $\mathbf{x} \notin \mathcal{D}'$, in which case $\epsilon \leq \mathbf{p}_{BNN}(y|\mathbf{x}) \leq 1 - \epsilon$, i.e. $\mathbf{x}$ is classified with low output probability and cannot be adversarial, or 2) $\mathbf{x} \in \mathcal{D}'$, in which case $\mathbf{x}$ is within some delta ball with centre $\mathbf{x}'$ and label $y' = 1$ or $y' = 0$. In the former case, $\mathbf{p}_{BNN}(y|\mathbf{x}) > 1 - \epsilon$ i.e. $\mathbf{x}$ is classified correctly with high output probability, and in the latter case $\mathbf{p}_{BNN}(y|\mathbf{x}) < \epsilon$, and $\mathbf{x}$ is still classified correctly with high output probability. Since every perturbed input $\mathbf{x}$ that is under a delta ball does not change the predicted class from that of the training example $\mathbf{x}'$, $\mathbf{x}$ cannot be adversarial either. □

Note that the assumption of lack of dataset ambiguity in the proof above can be relaxed, and the proof easily generalises to datasets with more than two classes. In §B we look at the proof critically, and in §C we discuss real-world conditions to replace the *idealised model structure* condition (first condition), suggesting that existing models with real data *do* capture sensible invariances from the dataset, enough to be regarded *empirically* as generalising well with *high coverage*. In §5 we relate such idealised BNNs to BNNs used in practice.

### 4.1 PROOF IMPLICATIONS

Our proof uses simple tools (continuity of NNs) as we do not need more elaborate techniques to demonstrate our claims. Despite the simplicity of the tools, the implications of the results above are profound. We suggested an explanation for *why* BNNs have been observed to be robust to adversarial examples, further suggesting a link between input-space density (which could be obtained from a generative model) and epistemic uncertainty obtained from discriminative probabilistic models. We showed that idealised BNNs *cannot have* adversarial examples, and (together with results presented in the next section) relate these to real-world conditions on BNNs. This observation allows us to focus research for robust machine learning towards tools which answer (or approximate) our conditions better, as discussed in §6. Further, we gave theoretical support for the hypothesis suggested in (Nguyen et al., 2015) (for the first time in the literature as far as we are aware), for which empirical support was given by Li (2018). Lastly, our proof highlights *sufficient* conditions for robustness which resolves the inconsistency between (Nguyen et al., 2015; Li, 2018) and (Gilmer et al., 2018) as discussed next.

### 4.2 ADVERSARIAL EXAMPLES ON THE SPHERES DATASET

Gilmer et al. (2018) constructed adversarial examples by constraining the perturbed example to lie on one of the two spheres the data was generated from, i.e. in high input density regions, demonstrating that it *is possible* to have adversarial examples lying in high density regions of the input space. This *stands in contrast* to the hypothesis suggested in (Nguyen et al., 2015), for which empirical evidence was collected by Li (2018). This rather puzzling observation can be explained using our results above, where we showed that adversarial examples *must exist only* in low density regions of the input space *as long as the model captures relevant data invariances*. I.e. when the model is built to capture the invariances in the data generating distribution then adversarial examples cannot

exist in high density regions. In the case of the spheres dataset, an idealised NN (a NN which is *rotation invariant*) will not have adversarial examples *on the spheres*, as discussed in §A.1. An idealised *BNN* will increase its uncertainty for off-manifold adversarial examples (since it never saw them before), and will thus reject all points off the spheres. Thus an idealised BNN is robust to adversarial examples with the spheres dataset. The results reported by Li (2018) can be explained by our observation in §C, suggesting that empirical low test error corresponds to high coverage.

### 4.3 GENERALISATION TO OTHER IDEALISED MODELS

Our proof trivially generalises to other idealised models that satisfy the two conditions set above (idealised architecture and idealised ability to indicate invalid inputs – definition 5 for the case of idealised BNN models). In appendix D we discuss which idealised models other than BNNs satisfy these two conditions, and further justify why we chose to continue our developments below empirically studying near-idealised BNNs.

## 5 EMPIRICAL EVIDENCE

In this section we give empirical evidence supporting the arguments above. We demonstrate the ideas using *near-perfect epistemic uncertainty obtained from HMC* (considered 'gold standard' for inference with BNNs (Neal, 1995)), and with *image data for which we know the ground-truth image-space density*. We give first experimental evidence in the literature (to the best of our knowledge) supporting the hypothesis that *ground truth image density* diminishes as images become adversarial, that uncertainty *correlates with input image density*, and that *state-of-the-art adversarial crafting techniques fail with HMC*. We then test how these *ideas transfer to non-idealised* data and models, demonstrating *failures of dropout uncertainty* on MNIST, and propose a *new attack* and a *mitigation* to this attack. We finish by assessing the robustness of our mitigation with a VGG13 variant, given in §G (due to space constraints).

### 5.1 IDEALISED CASE

In this subsection we are only concerned with 'near idealised' data and inference, assessing the definitions in the previous section. We start by deriving a new image dataset from MNIST (LeCun & Cortes, 1998), for which we know the ground truth density in the image space for each example $\mathbf{x}$, and are therefore able to determine how far away it is from the data distribution.

Our dataset, Manifold MNIST (MMNIST) was constructed as follows. We first trained a variational auto-encoder (VAE) on MNIST with a 2D latent space. We chose three image classes (0, 1, and 4), discarding the latents of all other classes, and put a small 'Gaussian bump' on each latent point from our 3 classes. Summing the bumps for each latent class we get an analytical density corresponding to this class. We then discarded the MNIST latents, and defined the mixture of the 3 analytical densities in latent space as our ground truth image density (each mixture component identified with its corresponding ground truth class). Generating 5,000 samples from this mixture and decoding each sample using our fixed VAE decoder, we obtained our training set for which each image has a ground truth density (Fig. 2, see appendix E for density calculation). Note that this dataset does not satisfy our lack-of-data-ambiguity assumption above, as seen in the figure.

First we show that the density decreases on average for image $\mathbf{x} \sim \texttt{MMNIST}$ as we make $\mathbf{x}$ adversarial (adding perturbations) using a *standard LeNet NN* classifier as implemented in Keras (LeCun et al., 1998; Chollet, 2015). Multiple images were sampled from our synthetic dataset, with the probability of an image in the input space plotted as it becomes adversarial for both targeted and non-targeted FGM (Goodfellow et al., 2014) attacks (Fig. 3). Together with Fig. 3, trajectories from the targeted attack (FGM) on MMNIST, seen in Fig. 12a in appendix F, show that even when the adversarial images still resemble the original images, they already have low probability under the dataset. Further, Fig. 12b shows that the deterministic NN accuracy on these images has fallen, i.e. the generated images successfully fool the model.

Next, we show that near-perfect epistemic uncertainty correlates to density under the image manifold. We use $\mathcal{D}_{\text{grid}}$ given by a grid of equally spaced poitns over the 2D latent space (Fig. 4). We used a BNN with LeNet architecture and HMC inference to estimate the epistemic uncertainty (Fig. 5, visualised in the VAE latent space; Shown in white is uncertainty, calculated by decoding each latent

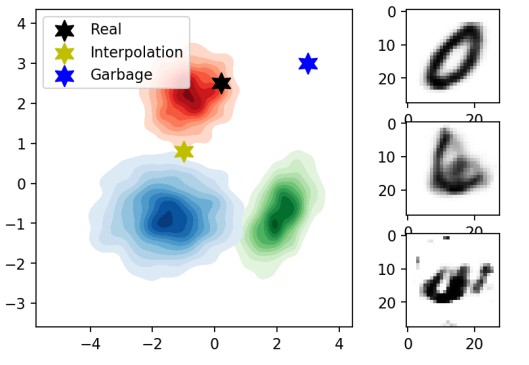

Figure 2: **Manifold MNIST ground-truth density in 2D latent space with decoded image-space realisations** (a real-looking digit (top), interpolation (middle), and garbage (bottom)).

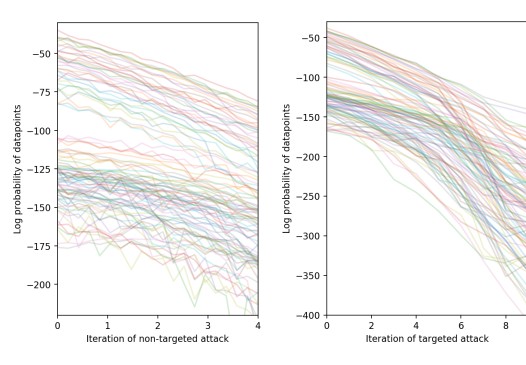

(a) Non-targeted  (b) Targeted

Figure 3: Ground-truth density v.s. step for FGM attacks on the decoded images with a deterministic classification NN. **Note the decreasing density as the images become adversarial**.

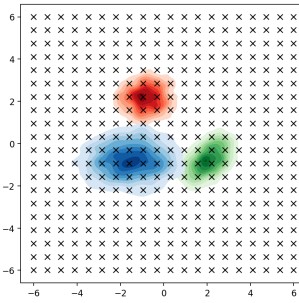

Figure 4: Our $\mathcal{D}_{grid}$ dataset depicted in 2D latent space with crosses overlaid ontop of Manifold MNIST.

Figure 5: Manifold MNIST 2D latent space with HMC MI projected from image space, showing "near perfect" uncertainty.

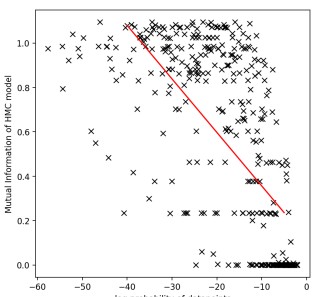

Figure 6: HMC MI v.s. log density of $\mathcal{D}_{grid}$ latent points. **Note the strong correlation between the density and HMC MI**.

point into image space, and evaluating the MI between the decoded image and the model parameters; A lighter background corresponds to higher uncertainty). In Fig. 6 we show that uncertainty correlates to density on the images from $\mathcal{D}_{grid}$.

Finally, we show that adversarial crafting fails for HMC. In this experiment we sample a new realisation from the HMC predictive distribution with every gradient calculation, in effect approximating the infinite ensemble defined by an idealised BNN. We used a non-targeted attack (MIM, first place in the NIPS 2017 competition for adversarial attacks (Dong et al., 2017)), which was shown to fool finite deterministic ensembles and be robust to gradient noise. Table 1 shows success rate in changing test image labels for HMC v.s. a deterministic NN, for maximum allowed input perturbation of sizes[4] $\epsilon \in \{0.1, 0.2\}$, v.s. a control experiment of simply adding noise of magnitude $\epsilon$. Also shown average image entropy. Note HMC BNN success rate for the attack is similar to that of the noise, v.s. Deterministic where random noise does not change prediction much, but a structured perturbation fools the model *very* quickly. Note further that HMC BNN's entropy increases quickly, showing that the model has many different possible output values for the perturbed images.

Table 1: MIM untargeted attack with two max. perturbation values $\epsilon$, applied to HMC BNN and Deterministic NN, showing success rate (*lower is better*) in changing true label for attack, changing true label by adding noise of same magnitude (control), and showing average entropy $\mathcal{H}$ for perturbed images. Shown mean $\pm$ std with 5 experiment repetitions.

|  | HMC BNN | | | | Deterministic NN | | | |
|---|---|---|---|---|---|---|---|---|
|  | Adv. succ. | Noise succ. | Adv. $\mathcal{H}$ | Noise $\mathcal{H}$ | Adv. succ. | Noise succ. | Adv. $\mathcal{H}$ | Noise $\mathcal{H}$ |
| $\epsilon = 0.1$ | $0.14 \pm 0.03$ | $0.10 \pm 0.01$ | $0.47 \pm 0.00$ | $0.33 \pm 0.00$ | $0.52 \pm 0.0$ | $0.03 \pm 0.001$ | $0.45 \pm 0$ | $0.06 \pm 0$ |
| $\epsilon = 0.2$ | $0.32 \pm 0.02$ | $0.23 \pm 0.01$ | $0.59 \pm 0.02$ | $0.53 \pm 0.01$ | $0.97 \pm 0.0$ | $0.03 \pm 0.002$ | $0.03 \pm 0$ | $0.08 \pm 0$ |

---

[4]Note that here we use $\epsilon$ to denote *maximum perturbation magnitude*, as is common in the literature, not to be confused with our $\epsilon$ from the proof.

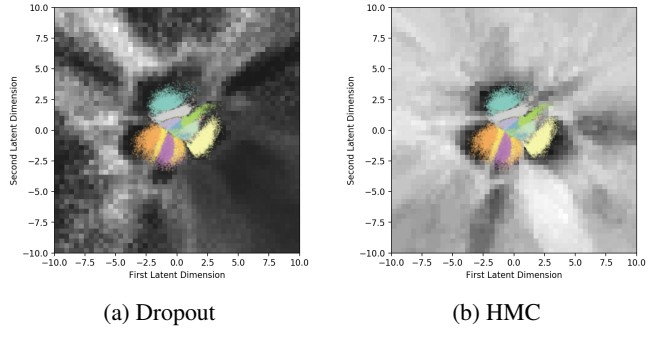

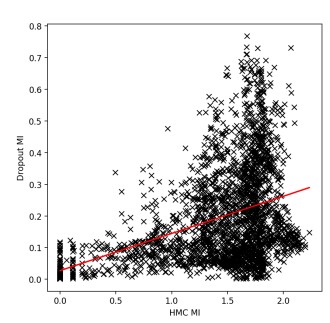

(a) Dropout                    (b) HMC

Figure 7: MNIST projected into 2D latent space with projected image-space MI for *dropout* and *HMC* inference. **Note the holes in uncertainty far from the data for dropout**.

Figure 8: Dropout MI v.s. HMC MI (note the correlation, but also that **dropout holes** lead to zero MI when HMC MI is non-zero).

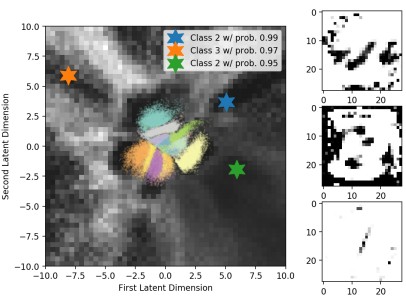

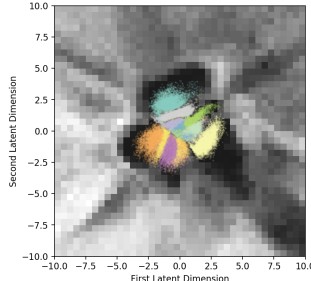

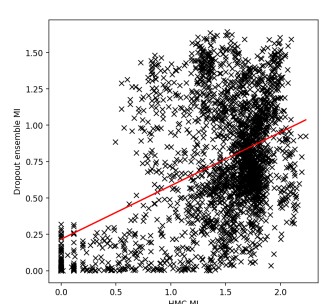

Figure 9: Three cherry-picked 'garbage' images classifying with high output prob., from 'holes' in dropout uncertainty over MNIST, and their locations in latent space.

Figure 10: 2D latent space with MI of *dropout ensemble* on MNIST, showing **fewer uncertainty 'holes'** v.s. dropout (7a).

Figure 11: Dropout ensemble MI v.s. HMC MI (most of the mass on the right has been **shifted up**, i.e. dropout holes are covered).

## 5.2 NON-IDEALISED CASE

Here we compare real-world inference (specifically, dropout) to near-perfect inference (HMC) on real noisy data (MNIST). We use the same encoder as in the previous section to visualise the model's epistemic uncertainty in 2D (Fig. 7). Note the dropout uncertainty 'holes' compared to HMC. We plot the dropout MI v.s. HMC MI for the grid of points $\mathcal{D}_{\text{grid}}$ as before in Fig. 8.

## 5.3 NEW ATTACK AND DEFENCE

We use the dropout failure case above to suggest a new attack generating 'garbage' images with high output probability, which does not require gradient information but instead queries the model for its confidence: First, collect a dataset of images, and project to 2D. Grid-up the latent space (Fig. 15 in appendix F) and query the model for uncertainty on each grid point. Order by distance from the nearest training point, and decode the farthest latents with low MI (i.e. points far from the training set on which the model is confident). Example crafted images given in Fig. 9. We further suggest a mitigation here, using intuition from above: we use an ensemble of randomly initialised dropout models (Fig. 10), and show that ensemble correlation with HMC MI fixes the uncertainty 'holes' to a certain extent (Fig. 11). In the appendix (F) we give quantitative results comparing the success rate of the new attack to FGM's success rate, and show that dropout ensemble is more robust to the state-of-the-art MIM attack compared to a single dropout model. We further show that the equivalent *Deterministic* model ensemble uncertainty contains more uncertainty 'holes' than the dropout ensemble.

## 6 DISCUSSION

Our result gives intuition into *why* dropout, a technique shown to relate to Bayesian modelling, seems to be effective in identifying adversarial examples. We presented several idealised models which satisfy the conditions we defined for robustness, opening the door for research into how various *practical* tools can approximate our *idealised* conditions. We highlighted that the main difficulty with modern BNNs is not coverage, but rather that approximate inference doesn't increase the un-

certainty fast enough with practical BNN tools (we show this in figures 7a, demonstrating that we have holes in the dropout uncertainty). In contrast, HMC (which is not scalable for practical applications) does not have such uncertainty holes, suggesting that we *must improve practical inference techniques in BNNs to improve robustness.*

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

# A    PROOF INTUITION

We informally discuss the sufficient conditions for robustness to adversarial examples in idealised models (informally, models with zero training loss). We give simple examples to depict the intuition behind these conditions. In the next section we formalise the conditions with a rigorous presentation and prove that under these conditions a model cannot have adversarial examples.

We need two key idealised properties to hold in order for a model not to have adversarial examples: idealised architecture (i.e. the model is invariant to all transformations the data distribution is invariant to), and ability to indicate when an input lies far from the valid input points (e.g. uncertainty is higher than some $\epsilon$, or the nearest neighbour is further than some $\delta$, in either case indicating 'don't know' by giving a low confidence prediction). The first property ensures the model has high *coverage*, i.e. generalises well to all inputs the data distribution defines as 'similar' to train points. The second property ensures the model can identify points which are far from all previously observed points (and any transformations of the points that the data distribution would regard as the same). Together, given a non-degenerate train set sampled from the data distribution, these two properties allow us to define an idealised model that would accepts and classify correctly all points one would define as a valid inputs to the model, and reject all other points.

The core idea of our proof is that a continuous classification model output doesn't change much within small enough neighbourhoods of points 'similar' to the training set points, at least not enough to change the training points' predictions by more than some $\epsilon$. A main challenge in carrying out a non-vacuous proof is to guarantee that such models generalise well, i.e. have high coverage. This is a crucial property, since many models are 'trivially' robust to adversarial examples by simply rejecting anything which is not identical to a previously observed training point. To carry out our proof we therefore *implicitly* augment the train set using all transformations $T \in \mathcal{T}$ extracted from the model and to which the model is invariant (and by the first condition, to which the data generating distribution is invariant). These transformations are implicitly extracted from the model architecture itself: For example, a translation invariant model will yield a train set augmented with translations. Thus the augmented train set might be infinite. We stress though that we do not change the train set for the *model training phase*; the augmented train set is only used to carry out the proof. In practice one builds the transformations the data distribution is invariant to into the model.

## A.1    MODEL INVARIANCES AND THE SPHERES DATASET

The implicitly augmented training set is used to avoid the degeneracy of the model predicting well on the train set but not generalising to unseen points. To gain more intuition into the role and construction of the set of transformations $\mathcal{T}$, recall the spheres dataset from (Gilmer et al., 2018), built of two concentric spheres each labelled with a different class. If it were possible to train a model *perfectly* with *all* sphere points, then the model could not have adversarial examples on the sphere because each point on the sphere must be classified with the correct sphere label. However it is impossible to define a loss over an infinite training set in practice, and a practical alternative to training the model with infinite training points is to build the invariances we have in the data distribution *into* our model. In the case of the spheres dataset we build a rotation invariance into the model. Since our model is now rotation invariant it is enough to have a single training point from each sphere in order for the model to generalise to the entire data distribution, therefore a model trained with only two data points will generalise well (have high coverage). A rotation invariant model trained with the two points is thus identical to an idealised model trained with the infinite number of points on the sphere. Formalising these ideas with the spheres example, in our proof below we rely on the implicitly constructed set of rotations $\mathcal{T}$; In the proof our train set (the two points) is augmented with the set of all rotations, thus yielding a set containing all points from the two spheres—in effect *implicitly constructing* an idealised model.

# B    PROOF CRITIQUE

We start by clarifying why we need to assume no ambiguity in the dataset. Simply put, if we had two pairs $(\mathbf{x}, 1)$ and $(\mathbf{x}, 0)$ for some $\mathbf{x}$ in our dataset, then no NN can be idealised following our definition (i.e. give probability 1 to the first observed point and probability 0 to the second). More generally, we want to avoid issues of low predictive probability near the training data; this assumption can be

relaxed assuming aleatoric noise and adapting the proof to use the mutual information rather than the entropy.

We use the idealised model architecture condition (and the set of transformations $\mathcal{T}$) to guarantee good coverage *in our proof*. CNNs (or capsules, etc.) capture the invariances we believe we have in our data generating distribution, which is the 'maxim' representation learning uses to generalise well. Note though that it might very well be that the model that we use in practice is not invariant to *all* transformations we would expect the data generating distribution to be invariant to. That would be a failure case leading to limited coverage; Compare to the spheres dataset example – if our model can't capture the rotation invariances then it might unjustifiably "reject" test points (i.e. classify them with low output probability, thus reduce coverage). In practice it is very difficult to define what transformations the data distribution is invariant to with real-world data distributions. However, we can estimate model coverage (to guarantee that the model generalises better than a look-up table or nearest neighbours) by empirical means as well. For example, we observe empirically on a variety of real-world tasks that CNNs have low uncertainty on test images which were sampled from the same data distribution as the train images, as we see in our experiments below and in other works (Kendall & Gal, 2017). In fact, there is a connection between a model's generalisation error and its invariance to transformations to which the data distribution is invariant, which we discuss further in appendix C. **This suggests that existing models with real data do capture sensible invariances from the dataset, enough to be regarded empirically as generalising well.**

Next we look at the proof above in a critical way. First, note that our argument does not claim the *existence* of an idealised BNN. Ours is not an 'existence' proof. Rather, we proved that under the definition above of an idealised BNN, such a BNN cannot have adversarial examples. The interesting question which follows is 'do there exist real-world BNNs and inference which approximately satisfy the definition?'. We attempt to answer this question empirically in the experiments section. Further note that our idealised BNN definition cannot hold for *all possible* BNN architectures. For a BNN to approximate our definition it has to increase its uncertainty fast enough. Empirically, for many practical BNN architectures the uncertainty indeed increases far from the data (Gal, 2016). For example, a single hidden layer BNN with sine activation functions converges to a GP with an RBF kernel as the number of BNN units increases (Gal & Turner, 2015); Both the RBF GP and the finite BNN possess the desired property of uncertainty increasing far from the training set (Gal & Turner, 2015). This property has also been observed to hold empirically for deep ReLU BNNs Gal (2016). In the same way that our results depend on the model architecture, not all GPs will be robust to adversarial examples either (e.g. a GP could increase uncertainty too slowly or not at all); This depends on the choice of kernel and kernel hyper-parameters. The requirement for the uncertainty to increase quickly enough within a region where the function does not change too quickly raises interesting questions about the relation between Lipschitz continuity and model uncertainty. We hypothesise that a relation could be established between the Lipschitz constant of the BNN and its uncertainty estimates.

Finally, our main claim in this work is that the *idealised* Bayesian equivalents of *some* of these other practical NN architectures will not have adversarial examples; In the experiments section we demonstrate that realistic BNN architectures (e.g. deep ReLU models for MNIST classification), with near-idealised inference, approximate the property of perfect uncertainty defined above, and further show that practical approximate inference such as dropout inference approximates some of the properties but fails for others.

## C  COVERAGE

To see why low test error implies high coverage, we present a simple argument that relies on the idealised case of zero expected error (error w.r.t. the data distribution) and the assumption that test error is representative of the expected error. Define a model $f(\mathbf{x})$ to be invariant to a transformation $T$ almost everywhere (a.e.) when $f(\mathbf{x}) = f(T(\mathbf{x}))$ for all $\mathbf{x} \in \mathcal{X}$ up to a zero measure set (i.e. for almost all $\mathbf{x} \in \mathcal{X}$); Assume that the data distribution has no ambiguity (i.e. a point $\mathbf{x}$ with non-zero probability for $y$ must have zero probability for $y' \neq y$). If the model $f(\mathbf{x})$ has zero *expected error* then $f(\mathbf{x}) = y$ a.e. in $\mathcal{X}$ with a corresponding $y$ having non-zero probability conditioned on $\mathbf{x}$, $p(y|\mathbf{x})$. For all transformations $T \in \mathcal{T}$ to which the data distribution is invariant, i.e. $p(y|\mathbf{x}) = p(y|T(\mathbf{x}))$, there exists that $y$ has non-zero probability conditioned on $T(\mathbf{x})$ as well, and therefore (from the lack of ambiguity assumption) it must hold that $f(T(\mathbf{x})) = y = f(\mathbf{x})$. Therefore the model $f(\mathbf{x})$ is invariant to $T$ a.e. as well.

Empirically, we observe near-idealised HMC LeNet BNN and dropout LeNet variants to have low test error with test points following the same distribution as the train points (we got ¿99% accuracy on the MNIST test set with a dropout BNN in our experiments). We use this as evidence towards the claim that our suggested models generalise well (have high coverage). Further, zero coverage for *invalid* inputs (i.e. the model saying 'don't know' for out-of-distribution examples, for example by giving uniform probabilities) is a desired property of our model.

Note that to get *full coverage* (i.e. not rejecting a single valid point) we must assume that for every valid input $\mathbf{x}'$ there exists some transformation $T \in \mathcal{T}$ mapping some training point $\mathbf{x}$ to $\mathbf{x}'$. This is more difficult to formalise for non-idealised models though.

## D  GENERALISATION TO OTHER IDEALISED MODELS

Here we discuss which idealised models satisfy our two conditions. The class of idealised models which satisfy our defined properties above is varied. The question of interest is what models are most suitable for which task, and which models approximate the idealised properties best.

We contrast several idealised models on the spheres dataset (Gilmer et al., 2018) to assess which could and could not satisfy our conditions. We use the spheres dataset here since we know the set of transformations an idealised model must be invariant to (i.e. if a model can satisfy the first condition). We will look at a NN with ReLU non-linearities (i.e. with no special invariances), a BNN with the same structure, a NN which is rotation invariant, a BNN which is rotation invariant, an RBF network (with either architecture), standard nearest neighbours, and nearest neighbours in feature space with some deterministic feature extractor, all using finite training sets.

1. A NN with no invariances can have adversarial examples (as demonstrated in (Gilmer et al., 2018)).

2. A NN with rotation invariances will not have adversarial examples *on the spheres* (following our argument in section A). However, the NN *might have* 'garbage' adversarial examples which classify with high output probability far away from the data, where the model might be wrongly confident (we show this in our experiments below as well). An idealised NN can predict with arbitrary high output probability far away from the training set, and there is no way to enforce the model not to do so – we can't iterate over all points 'not in the train set' and force a standard deep NN model to predict with near uniform output probability on these points.

3. A BNN with no invariances *cannot have adversarial examples* on the sphere but will have low coverage. It will output a prediction for the finite train set points, and will output 'don't know' (i.e. near-uniform probability) for all other points on the sphere which it didn't have in the train set. The BNN *will not have* garbage adversarial examples far from the data (since the model averages many different functions, each giving different values far from the data, in effect increasing its uncertainty and pushing the predictive probability to uniform).

4. A BNN with rotation invariances will *have no adversarial examples* on the sphere (following the arguments in points 2 and 3) and with *full coverage* for all sphere points. It will have no garbage adversarial examples (following the argument in point 3). As mentioned we still assume a finite training set for the BNN, but having the model rotation invariant makes this equivalent to the case in point 3 with an infinite train set which includes all sphere points.

5. An idealised RBF network which collapses to uniform prediction fast enough follows the same intuition of idealised BNNs above – both for a rotation invariant RBF network as well as for a model with no invariances.

6. Nearest neighbour which uses thresholding to declare 'don't know' and with a finite dataset (again, all above also used finite dataset and invariances built into the model itself) will have no adversarial examples, but will have low coverage following the same arguments in point 3. The only way to fix the issue of low coverage with standard nearest neighbours is to explicitly assume that the model is trained with an infinite training set with all sphere points (in which case it will have proper coverage). Note the difference to the BNN / NN models which use a finite training set with invariances built into the model to implicitly augment the train set. A possible way to alleviate this issue is to perform nearest neighbour

in feature space, building invariances into the distances nearest neighbours uses, allowing a finite training set to be used to get full coverage. This idea is developed further in (Papernot & McDaniel, 2018).

Probability thresholding using the Bayesian approach plays an important role in our proof, but is not the only way to declare 'don't know' as we saw above. Note though that nearest neighbour thresholding is not trivial: Even though one might define for example 'distance in input space to nearest neighbour' in order to declare an output as 'don't know', in practice thresholding the distance in input space (or, for that matter, in feature space) can affect points differently in different parts of the input space (e.g. we want to have low threshold in high density regions v.s. high threshold in low density regions). The Bayesian approach gives tools to do the thresholding in the output probability space, further allowing us to define a tolerance to false positives if our uncertainty is calibrated. Lastly, we note that we can't simply define a third class (class 2) to indicate 'don't know', even in the rotation invariant NN. The quotient group of all distinct points in our spheres dataset (after identifying all points on the surface of a sphere with some radius $r$ as identical to each other) is still infinite: It is all the non-negative reals (corresponding to sphere radii). Out of these, one point (r=1) corresponds to class 1, one point (r=1.3) corresponds to class 0. In this case there exist infinitely many points that will be assigned class 2. The point of the augmented train set trick from the proof is to induce finite train sets over which we can define the invariant model loss (which is feasible in practice). But with the 'don't know' class the train set (in either case) is infinite, which means it is infeasible to define a loss over it. For these reasons we chose to continue our developments studying idealised and near-idealised BNNs.

## E    IMAGE DENSITY CALCULATION

For our MMNIST dataset we have an analytical expression for the density in the latent space for each class $c$: a Gaussian $p(\mathbf{z}|c)$ (with $\mathbf{z}$ the latent variable). With this density we can calculate the density of an observed image $\mathbf{x}$ by MC integration over the latent space:

$$p(\mathbf{x}) = \int p(\mathbf{x}|\mathbf{z})p(\mathbf{z}|c)p(c)\mathrm{d}\mathbf{z}\mathrm{d}c \approx \frac{1}{T}\sum_{t=1}^{T} p(\mathbf{x}|\widehat{\mathbf{z}}_t)$$

with $\widehat{\mathbf{z}}_t \sim p(\mathbf{z}|\widehat{c}_t)$ and $\widehat{c}_t \sim p(c)$. In practice we use importance sampling for density calculations. In Fig. 13 we show that the ground truth latent space density correlates strongly with the image density obtained from this estimator on test MMNIST images.

## F    MORE EMPIRICAL RESULTS

We next assess the success rate of getting garbage images which classify with high output probability ($> 0.9$), comparing our new latent space attack which does not use gradient information to the untargeted FGS attack (which does use gradients), on a dropout NN with MNIST. Note though that the sample size used here is rather small (15 generated images).

Success rate (*higher is better for attack*) generating garbage examples which classify with probability greater than 0.9:

| Attack | Success rate |
|---|---|
| Latent space attack | 0.76 |
| Untargeted FGM | 0.60 |

Next we show the robustness of dropout ensemble v.s. dropout with a MIM attack on MNIST; Note the improved robustness for the ensemble.

Success rate (*lower is better for defence*) in changing image label with the MIM attack on both dropout and dropout ensemble (4 dropout models). Shown are mean and std with 5 experiment repetitions:

| Perturbation magnitude | Dropout | Dropout ensemble |
|---|---|---|
| $\epsilon = 0.1$ | $0.37 \pm 0.02$ | $0.26 \pm 0.00$ |
| $\epsilon = 0.175$ | $0.91 \pm 0.00$ | $0.79 \pm 0.02$ |

Further figures (referenced from the main text) are given below.

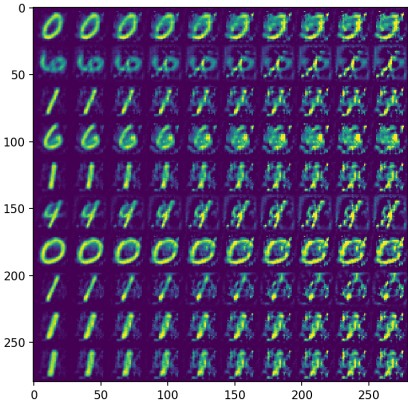

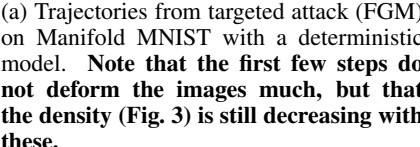

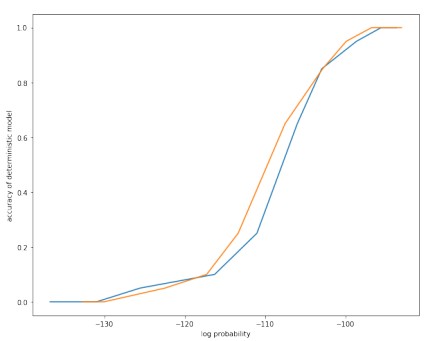

(a) Trajectories from targeted attack (FGM) on Manifold MNIST with a deterministic model. **Note that the first few steps do not deform the images much, but that the density (Fig. 3) is still decreasing with these.**

(b) Prediction accuracy v.s. image density on the trajectory images from the targeted attack (FGM), with Manifold MNIST and a deterministic model. **Note that the accuracy diminishes as the image density becomes smaller, i.e. as the images become adversarial; In other words, the attack is** *successful.*

Figure 12: Image trajectories and their accuracy v.s. density.

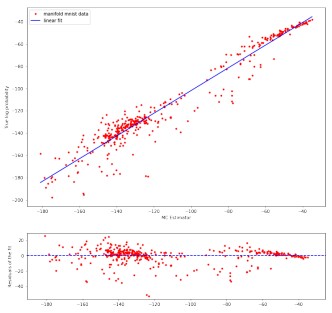

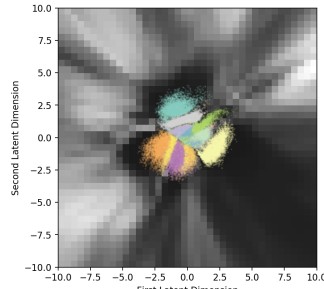

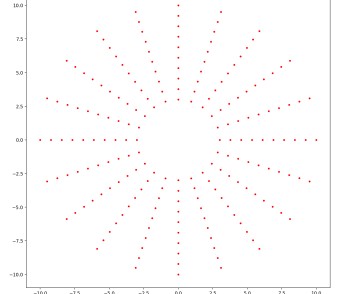

Figure 13: Ground truth latent log probability v.s. image density obtained through importance sampling (§E) for test images from the MMNIST dataset.

Figure 14: 2D latent plot of deterministic NN MI. Note the large gaps in uncertainty (larger than dropout ensemble's gaps).

Figure 15: New attack; This figure depicts our grid over latent space to look for uncertainty 'holes' far from the data.

## G  REAL-WORLD CATS VS DOGS CLASSIFICATION

We extend the results above and show that an ensemble of dropout models is more robust than a single dropout model using a VGG13 (Simonyan & Zisserman, 2015) variant on the ASIRRA (Elson et al., 2007) cats and dogs classification dataset. We retrained a VGG13 variant ((Simonyan & Zisserman, 2015), with a reduced number of FC units) on the ASIRRA (Elson et al., 2007) cats and dogs classification dataset, with Concrete dropout (Gal et al., 2017) layers added before every convolution. We compared the robustness of a single Concrete dropout model to that of an ensemble following the experiment setup of (Smith & Gal, 2018). Here we used the FGM attack with $\epsilon = 0.2$ and infinity norm. Example adversarial images are shown in Fig. 16. Table 2 shows the AUC of

different MI thresholds for declaring 'this is an adversarial example!', for all images, as well as for successfully perturbed images only (S). Full ROC plots are given in Fig. 17. We note that the more powerful attacks succeed in fooling this VGG13 model, whereas *dropout Resnet-50* based models seem to be more robust (Smith & Gal, 2018). We leave the study of model architecture effect on uncertainty and robustness for future research.

Table 2: AUC for MIM attack on VGG13 models (*higher is better*), trained on real-world cats v.s. dogs classification, for both a single Concrete dropout model, as well as for an ensemble of 5 Concrete dropout models

| Model | AUC | AUC (S) |
|---|---|---|
| Concrete Dropout | 0.63 | 0.61 |
| Concrete Dropout Ensemble | 0.77 | 0.74 |

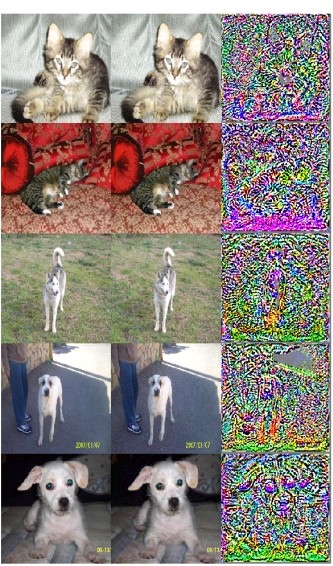

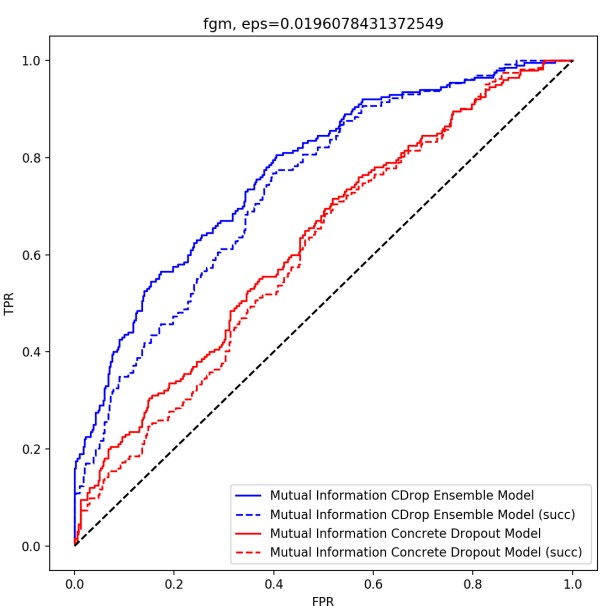

Figure 16: Example dataset images, generated adversarial counterparts, and the perturbation.

Figure 17: ROC plot of dropout and dropout ensemble using MI thresholding to declare 'adversarial', evaluated both on all examples, and on successfully perturbed examples (marked with 'succ').

