# OpenReview forum: "Sufficient Conditions for Robustness to Adversarial Examples: a Theoretical and Empirical Study with Bayesian Neural Networks"
_ICLR.cc/2019/Conference_

### Official Review · AnonReviewer3 · 2018-10-30
**Results seem to be shallow and vague**

**Rating:** 4
**Confidence:** 4

**Review:**

This paper extends the definition of adversarial examples to the ones that are “far” from the training data, and provides two conditions that are sufficient to guarantee the non-existence of adversarial examples. The core idea of the paper is using the epistemic uncertainty, that is the mutual information measuring the reduction of the uncertainty given an observation of the data, to detect such faraway data. The authors provided simulation studies to support their arguments.

It is interesting to connect robustness with BNN. Using the mutual information to detect the “faraway” datapoint is also interesting. But I have some concerns about the significance of the paper:
1.  The investigation of this paper seems shallow and vague.
    (1). Overall, I don’t see the investigation on the “typical” definition of adversarial examples. The focus of the paper is rather on detecting “faraway” data points. The nearby perturbation part is taken care by the concept of “all possible transformations” which is actually vague.
    (2). Theorem 1 is basically repeating the definition of adversarial examples. The conditions in the theorem hardly have practical guidance: while they are sufficient conditions, all transformations etc.. seem far from being necessary conditions, which raises the question of why this theory is useful? Also how practical for the notion of “idealized NN”?
    (3). What about the neighbourhood around the true data manifold? How would the model succeed to generalize to the true data manifold, yet fail to generalize to the neighbourhood of the manifold in the space?  Delta ball is not very relevant to the “typical” definition of adversarial examples, as we have no control on \delta at all.
2. While the simulations support the concepts in section 4, it is quite far from the real data with the “typical” adversarial examples.

I also find it difficult to follow the exact trend of the paper, maybe due to my lack of background in bayesian models.
1. In the second paragraph of section 3, how is the Gaussian processes and its relation to BNN contributing to the results of this paper?
2. What is the rigorous definition for \eta in definition 1?
3. What is the role of $\mathcal{T}$, all the transformations $T$ that introduce no ambiguity, in Theorem 1. Why this condition is important/essential here?
4. What is the D in the paragraph right after Definition 4? What is D’ in Theorem 1?
5. Section references need to be fixed.

---

> ### Author Response · Authors · 2018-11-08
> **Response to reviewer 2**
>
> We thank the reviewer for spending time reading our paper. The reviewer seems to have misunderstood large portions of our submission, and we would like to clarify some of these misunderstood points.
>
> * The reviewer wrongly characterises the paper as focusing on ‘faraway’ or garbage examples only, rather than adversarial examples generated by nearby perturbations ("Overall, I don’t see the investigation on the “typical” definition of adversarial examples"; "The focus of the paper is rather on detecting “faraway” data points."). This is incorrect (see Definition 1 and Theorem 1). We think the confusion might have been caused because the reviewer misunderstood our notation x + η (a standard notation in the field for a small perturbation η around x, see eg (Papernot et al., 2016; Goodfellow et al., 2014), and our Definition 1 which states “small perturbation η”). Theorem one clearly deals with the case of x + η as well as “garbage points”.
>
> * The reviewer writes ‘the nearby perturbation part is taken care of by the concept of all possible transformations’. This is not what we intended. Rather, the concept of the invariances of the dataset is included in order to give the idealised classifier non-vacous coverage. Without this condition, it is possible for an idealised model to “reject all points it has never seen before” as adversarial. For example, consider a classifier that keeps the dataset {x_i, y_i} in memory, and classifies new points x by returning y_i with probability 1 if there exists an i such that x_i = x, and returns a uniform probability vector otherwise. This satisfies our definition of ‘high uncertainty away from the training data’ but is clearly not very useful. Introducing the class of transformations gives the idealised classifier *in the proof* the ability to generalise in a non-trivial way, though it is clearly a very strong assumption that does not hold in real networks. This assumption, as well as real-world alternatives to high coverage, is discussed in Appendix C.
>
> * The reviewers second point is that our conditions are very strong, and it is not clear they are necessary. We do not claim that the conditions are *necessary* -  our results are giving *sufficient* conditions for robustness (as our paper title says). Conditions do not need to be necessary for them to be useful. We think that our conditions provide a useful framework to set directions for future work. We would point out that as far as we know there are no other attempts to formalise what an idealised model without adversarial examples would be like, and we think it is fair to say that empirical attempts to find sufficient conditions by producing a model which does not exhibit adversarial examples have been unsuccessful as of yet.
>
> * We are unsure what the reviewer means by ‘generalising to the neighbourhood of the true data manifold’. If by the true data manifold they mean the support of P(x), then ‘generalising’ outside this region is more or less meaningless. This is mentioned in the related work section about oracles - “can a point outside the true data distribution be assigned a label in a meaningful way?”. We would also highlight that the delta balls are a subset of the neighbourhood around the true data manifold.
>
> * Lastly, the simulations are not designed to resemble real data necessarily, but to show under some idealised conditions (but less idealised than used in the proof) that the properties described do indeed approximately hold for realisable networks. Indeed, one of the experiments (on manifold mnist) is designed to address the previous point about the data support, by showing that adversarial examples are being ‘moved away’ from the data distribution. In Appendix G we give results on real world cats-vs-dogs classification as well.
>
> --
> To address the questions the reviewer had about the trend of the paper :
> * GPs are the infinite limit of BNNs, and share many properties with them (Matthews et al., 2017). The point of the reference to GPs is that these are an example of a model for which the uncertainty is easy to evaluate. This is then used to illustrate what we hope to obtain in the case of idealised neural networks (gp-like uncertainty).
> * Eta is a vector in R^dim, the image space, where we follow standard notation as mentioned above. There is no commonly agreed on definition of adversarial examples more rigorous than the one we provide. However, the argument in the paper does not rely on any properties of eta other than there is *some* eta we consider ‘small enough’ to be adversarial.
> * As mentioned above, this condition is essential to avoid a classifier that is uncertain everywhere from satisfying our definition of an idealised model.
> * D refers to the training set, which is mentioned above equation 1. D’ refers to the union of delta balls around the training set, which is in definition 5.
> * Thanks for pointing this out, we will fix this in our next draft.

---

> > ### Author Response · Authors · 2018-11-09
> > **Question to reviewer**
> >
> > Could we ask if the above clarified the misunderstandings the reviewer had? We are happy to give further clarifications if the reviewer has further questions

---

> > > ### Comment · AnonReviewer3 · 2018-11-19
> > > **Sorry for the late reply, and thanks for the patient clarification.**
> > >
> > > I am sorry for the confusion in my review. I indeed misunderstood the \eta in definition 1, where originally I thought it was decided by Lemma 1, and the role of the invariance under all transformations. Thanks for the clarification.
> > >
> > > I think my evaluation still holds. To avoid other possible misunderstandings, let me summarize the theoretical result of the paper here: The main result of the paper is Theorem 1, which basically says under the following 3 conditions, classifier f has no adversarial examples.
> > > (1) The training set X can present the data manifold (under all the transformations that  maps);
> > > (2) f has 0 training error;
> > > (3) f has low output probability on D’.
> > > Based on Definition 1, the definition of adversarial examples, this theorem directly follows. (this point is also mentioned by Reviewer #2.)
> > > I think it is fair to say that the technical contribution is weak. Therefore, a key point of the review would be the contribution in fundamental understanding, and the practical guidance of the theory.
> > >
> > > Fundamental understanding:
> > > 1. I think it is interesting to connect bayesian networks with robustness. Introducing randomness has been shown to be helpful, and the “epistemic uncertainty” could be one of the perspectives.
> > > 2. However, theorem 1 actually gives pretty strong assumptions. For example, the training set should be able to cover the data manifold to guarantee the generalization on clean data. Arguably it is easy to provide many “sufficient” conditions, and what differs them is how natural/weak these conditions are (Of course, sufficient and necessary conditions would be ideal). This is essential for the meaningfulness/importance of a theorem. The contribution of the current result, in my opinion, is solely connecting bayesian networks with the randomness in achieving robustness, which seems to be lower than the ICLR bar.
> > > 3. It is argued in the paper that this theory provides an explanation for the dropout inference. But on the other hand, as mentioned above, its contribution in fundamental understanding seems incremental. It is also not clear why this explanation is the correct one, or better than other alternatives, for example like high frequency signals. (The high frequency view may arguably connect to the bayesian view here too, which in fact hurt the novelty of the results in this paper.)
> > > 4. In the rebuttal, “our conditions provide a useful framework to set directions for future work. We would point out that as far as we know there are no other attempts to formalise what an idealised model without adversarial examples would be like,… .” In general I don’t see why this framework is useful. (See the comments above.) It is easy to provide sufficient conditions, e.g. separable data with large margins.
> > > 5. I didn’t read the appendix for my original review. Appendix C has several discussions about these conditions, which in my opinion are interesting and important, and should be in the main content. I think the presentation of the paper needs to be significantly improved.
> > >
> > > Practical guidance: Given the strong conditions in theorem 1, it is hard to see how it can guide practical algorithms. Results in section 5 is only on synthesis data, while in section G, the idea of ensembling is not really new in the literature. It would have justified this point much more solidly, if the theorem could hint a new algorithm that empirically improve the results on some popular benchmarks . The essential difficulty in following the conditions here is how to achieve a model that would decreases its output probability slowly, while increases its uncertainty fast. It is also discussed in Appendix B (Again I think this part is also interesting and should be in the main content), but deferred as a hypothesis about Lipschitz constant without definitive answer. In fact, I think the modulus of continuity is more relevant here. But again, solely by itself such conceptual extension is not really useful.

---

> > > > ### Comment · AnonReviewer3 · 2018-11-19
> > > > **Con't**
> > > >
> > > > Lastly, I would like to emphasize that the presentation of this paper needs to be improved.
> > > > 1. For example,  the “small” \eta and “high” probability should be rigorous to make definition 1 a definition.
> > > > 2. In definition 2, what is the support of p, and what are the domain and codomain of T? Is the set of all the such transformations \mathcal{T} well defined?
> > > > 3. In the paragraph after definition 3, “(X,Y)” is used to represent the training set. Later in the paragraph after definition 4, “D” is used to represent the training set.
> > > > 4. In the paragraph after definition 4, is the H(p) here conditioned on the training set?
> > > > 5. The definition of the \delta ball is actually in a lemma, Lemma 1.
> > > > 6. For me, Appendix B,C are more important than the synthesis experiments in section 5. Also the verbose explanation before theorem 1 could also be improved.
> > > >
> > > > In summary, I think it is interesting to connect Bayesian networks and robustness in this paper. However, the investigation in the paper is a bit shallow, and it is not well justified how significant this contribution is neither in fundamental understanding or practical guidance. I would be happy to discuss further about the concerns I have above.
> > > >
> > > > Question: In section D, is it assumed that BNN has the property that the uncertainty increases fast away from the training set?

---

> > > > > ### Author Response · Authors · 2018-11-24
> > > > > **response**
> > > > >
> > > > > We thank the reviewer for their feedback and for taking part in an active discussion with us, and are glad that they found the discussion in the appendices helpful. This feedback on the structure and presentation of the paper has been fairly consistent among the reviewers, and we would appreciate the reviewer’s specific feedback on where they think they got confused with the contribution of the paper being otherwise than stated by our title and abstract (and introduction). We would strive to update and clarify this before the end of the revision period.
> > > > >
> > > > > Answering some of the reviewer’s questions and comments:
> > > > >
> > > > > * We would argue that the connection to randomisation would in fact be bad. As mentioned by reviewer 1, ‘mere’ randomisation of networks can provide a false sense of security. We would argue that the important part of the randomisation is performing approximate inference. Randomised models and Bayesian models are not equivalent - if we were able to evaluate the integral over the posterior in closed form as we can for a GP, then presumably this would be far better than a monte carlo approximation in terms of robustness to adversarial examples.
> > > > >
> > > > > * We are not sure what the reviewer is referring to by the ‘high frequency signals’ alternative. We assume the reviewer is referring to the theory that adversarial examples are due to models fitting high frequency noise in the dataset. This may very well also be true (it is likely that adversarial examples do not have a single cause).
> > > > >
> > > > > * The example that separable data with high margins would also not have adversarial examples is probably accurate, but this is a sufficient condition on the data, which we would argue is less interesting that sufficient conditions on the kind of model. After all, we do not have the luxury of choosing the data that we work with.
> > > > >
> > > > > * Lastly, we are glad that the reviewer found the appendices helpful, and we will bear this in mind in future work. It is unfortunate that these could not all be incorporated into the main body of the paper under the page constraint without making other parts less clear.

---

> > > > > > ### Author Response · Authors · 2018-11-24
> > > > > > **continued response**
> > > > > >
> > > > > > Let us also briefly answer the reviewers second question
> > > > > > 1.The reviewer comments that we should make the definitions ‘small’ and ‘high’ precise. These are indeed not particulary well defined, but we do not really use the property that eta is small or the probability is high in the proof - this is more to accord with the ‘common sense’ definition of an adversarial example, where small means ‘small enough i consider it adversarial’. In the general case, we think it would be difficult to define this in a sensible way (would the argument substantially change if we said ‘we consider examples with eta < 0.3243 adversarial’?
> > > > > >
> > > > > > 2. In this case, P(x,y) is the joint distribution of the data (we assume iid sampling), so its support is (some subset of) the space X x Y. The transformations are transformations in the input space, so they have domain/codomain t : X -> X. We thought that this was fairly obvious from common convention and the context in which t(x) is used, but we could clarify in the future.
> > > > > > Sorry for this inconsistency. However, both of X,Y and D for the dataset is pretty standard notation.
> > > > > >
> > > > > > 4. Yes - this is the predictive entropy of a model, so it is conditioned on the dataset
> > > > > > 5. Does the reviewer mean the definition of the Euclidean ball? This is indeed clarified in a footnote to avoid ambiguity in lemma one, but a 'delta ball' is just a way of saying 'a euclidean ball of radius delta', so we did not think this merited its own definition.
> > > > > > 6. We will bear this feedback in mind for future drafts

---

> > > > > > ### Comment · AnonReviewer3 · 2018-11-26
> > > > > > **Thanks for the reply**
> > > > > >
> > > > > > “We would argue that the connection to randomisation would in fact be bad.” Not sure if I understand this part correctly. I would argue that this is actually an evaluation problem. Overall, I don’t see any evidence that randomized models are less robust. More importantly, I was raising this point to evaluate the contribution of this paper to the fundamental understanding. The “randomisation” mentioned here should be taken as a general term. If the authors think the importance of performing approximate inference is the main conceptual contribution of the paper, then it should be emphasized and reflected in the paper. I don’t see it from the current version of the paper.
> > > > > >
> > > > > > I mentioned the ‘high frequency signals’ to provide an alternative explanation for why dropout would help in defence (dropout as a way in approximately removing high frequent signals). So to me, “dropout as performing approximate inference” proposed in this paper is not the unique possibility. What makes this explanation/hypothesis significant (more significantly compared to other alternatives)?
> > > > > >
> > > > > > Arguably, the conditions in this paper also depend on the data distribution. The first condition that the training set can represent the real data manifold is basically an implicit  assumption (if ). Also, if the manifolds of two different classes are close to each other, then a “no-adversarial-example” model would have to make lots of low-output-probability predictions, even for the real data. (Imagine x1 and x2 close to each other, both real data but with different true class labels.) This phenomenon traces back to the \eta in definition 1. Again, definition 1 is not rigorous, as it is missing a rigorous definition for \eta here (which should depend on the data distribution).

---

> > > > > > > ### Author Response · Authors · 2018-11-26
> > > > > > > **further reply**
> > > > > > >
> > > > > > > On randomisation, sorry for not being clear. I didn't mean that randomised model s are less robust - here, we meant to refer to the phenomenon that randomised models can provide a false sense of security because they randomise the gradients, which makes life harder for adversarial image generation algorithms that use gradient based optimisation, but don't really get rid of the problem in the sense that the adversarial regions of data space remain. The reviewers characterisation of this as an evaluation problem is also a fair characterisation, in the sense that we have to resort to randomisation to evaluate integrals approximately.
> > > > > > > There are, indeed, other explanations of what dropout does that are potentially useful. Sensitivity to high frequency signals is a possible cause of adversarial behaviour in image models, and one can see dropout as a regularisation technique that reduces this. However, we do not think this would be sufficient- one can fairly easily conceive of situations in which a model that only fits low frequency functions would have adversarial examples (there is an interesting discussion of this in https://arxiv.org/abs/1806.08734 - see figure 2. for an example.). It could certainly have 'garbage' examples far from all data on which it was very confident.
> > > > > > > It is a reasonable point that we do have some assumptions on the data, but we think they are weaker than 'separable with large margins'.
> > > > > > >
> > > > > > > This second point is interesting. This is related to the distinction in the paper between epistemic and aleatoric uncertainty. This situation corresponds to high aleatoric uncertainty (i.e the data is intrinsically noisy), or possibly to very fine distinctions in the data. This actually exists in some datasets - for example, on mnist, it is very easy to change a 7 into a 9 with a relatively small perturbation. However, this is of course not an 'adversarial' perturbation - we have really changed the data class. In this sense, eta does depend on the data - it must be 'small enough' to not change the class. However, we do not think that the property of eta is used anywhere in the argument. (i.e if it is smaller than delta it doesn't change the class prediction, and if it is larger then the uncertainty will be high). However, clearly this could be explained more clearly, and we will work on this in future drafts
> > > > > > > The distinction between epistemic and aleatoric uncertainty is discussed in the paper in more detail in section 3 - in particular see figure 1. The epistemic uncertainty should be low in the situation described here, where we are really changing the class in question

---

### Official Review · AnonReviewer1 · 2018-11-02
**Interesting and important theory, with questions about usefulness**

**Rating:** 5
**Confidence:** 3

**Review:**

In this paper, the authors posit a class of discriminative Bayesian classifiers that, under sufficient conditions, do not have any adversarial examples. They distinguish between two sources of uncertainty (epistemic and aleatoric), and show that mutual information is a measure of epistemic uncertainty (essentially uncertainty due to missing regions of the input space). They then define an idealised Bayesian Neural Network (BNN), which is essentially a BNN that 1) outputs the correct class probability (and always with probability 1.0) for each input in the training set (and for each transformation of training inputs that the data distribution is invariant under), and 2) outputs a sufficiently high uncertainty for each input not in the union of delta-balls surrounding the training set points. Similarly, an example is defined to be adversarial if it has two characteristics: it 1) lies far from the training data but is classified with high output probability, and it 2) is classified with high output probability although it lies very close to another example that is classified with high output probability for the other class. Condition 1) of an idealised BNN prevents Definition 2) of an adversarial example using the fact that BNNs are continuous, and Condition 2) prevents Definition 1) of an adversarial example since it will prevent "garbage" examples by predicting with high uncertainty (of course I'm glossing over many important technical details, but these are the main ideas if I understand correctly).

The authors backed up their theoretical findings with empirical experiments. In the synthetic MNIST examples, the authors show that adversarial attacks are indeed correlated with lower true input probability. They show that training with HMC results in high uncertainty for inputs not near the input-space, a quality certainly not shared with all other deep models (and another reason that Bayesian models should be preferred for preventing adversarial attacks). On the other hand, the out-of-sample uncertainty for training with dropout is not sufficient to prevent adversarial attacks, although the authors posit a form of dropout ensemble training to help prevent these vulnerabilities.

The authors are tackling an important issue with theoretical and technical tools that are not used often enough in machine learning research. Much of the literature on adversarial attacks is focused on finding adversarial examples, without trying to find a unifying theory for why they work. They do a very solid exposition of previous work, and one of the strengths of this paper comes in presenting their findings in the context of previously discovered adversarial attacks, in particular that of the spheres data set.

Ultimately, I'm not convinced of the usefulness of their theoretical findings. In particular, the assumption that the model is invariant to all transformations that the data distribution is invariant under is an unprovable assumption that can expose many real-world vulnerabilities. This is the case of the spheres data set without a rotation invariance from Gilmer et al. (2018). In the appendix, the authors mention that the data invariance property is key for making the proof non-vacuous, and I would agree. Without the data invariance property, the proof mainly relies on the fact that BNNs are continuous. The experiments are promising in support of the theory, but they do not seem to address this data invariance property. Indeed a model can prevent adversarial examples by predicting high uncertainty for all points that are not near the training examples, which Bayesian models are well equipped to do.

I also thought the paper was unclear at times. It is not easy to write such a technical and theoretical paper and to clearly convey all the main points, but I think the paper would've benefited from more clarity. For example, the notation is overloaded in a way that made it difficult to understand the main proofs, such as not clearly explaining what is meant by I(w ; p) and not contrasting between the binary entropy function used for the entropy of a constant epsilon and the more general random variable entropy function. In contrast, I thought the appendices were clearer and helpful for explaining the main ideas. Additionally, Lemma 1 follows trivially from continuity of a BNN. Perhaps removing this and being clearer with notation would've allowed for more room to be clearer for the proof of Theorem 1.

A more minor point that I think would be interesting is comparing training with HMC to training with variational inference. Do the holes that come from training with dropout still exist for VI? VI could certainly scale in a way that HMC could not, which perhaps would make the results more applicable.

Overall, this is a promising theoretical paper although I'm not currently convinced of the real-world applications beyond the somewhat small examples in the experiments section.

PROS
-Importance of the issue
-Exposition and relation to previous work
-Experimental results (although these were for smaller data sets)
-Appendices really helped aid the understanding

CONS
-Real world usefulness
-Clarity

---

> ### Author Response · Authors · 2018-11-08
> **response to reviewer 1**
>
> We thank the reviewer for the detailed feedback and for acknowledging that we tackled an important issue with theoretical and technical tools that are not used often enough in machine learning. The reviewer’s summary of our theoretical development is faithful, but we would like to address the reviewer’s comments on impact and real-world usefulness. This is followed by detailed answers to the minor comments and suggestions.
> --
> Major points:
> First, for the first time in the literature (as far as we know), we studied in detail a proposed mechanism to explain why BNNs have been observed to be empirically more robust to adversarial examples (with such empirical observations given by (Li & Gal, 2017; Feinman et al., 2017; Rawat et al., 2017; Carlini & Wagner, 2017)). We argue that idealised BNNs cannot have adversarial examples, and related these to real-world conditions on BNNs. More specifically, see the sequence of experiments starting with idealised models with idealised data (section 5.1), going through idealised models with real world data (table 1), and finishing with real world data and inference (dropout approx inference, section 5.2, as well as appendix G - REAL-WORLD CATS VS DOGS CLASSIFICATION).
>
> Secondly, for the first time in the literature, we proved (and gave empirical evidence in Fig 3) that a connection exists between epistemic uncertainty/input density and adversarial examples, a link which was only *hypothesised* previously (Nguyen et al., 2015). Further, our proof highlighted sufficient conditions for robustness which resolved the inconsistency between (Nguyen et al., 2015; Li, 2018) and (Gilmer et al., 2018).
>
> With regards to the the assumption that “the model is invariant to all transformations that the data distribution is invariant under” not being addressed: this assumption was used in the formal proof to get non-vacuous coverage. In appendix C we give a real-world alternative to this assumption, and in our experiments we demonstrate high coverage empirically (since this property is mostly impossible to assess otherwise, as the reviewer mentioned).
>
> Apart from resolving an open dispute in the literature, our observations above also suggest future research directions towards tools which answer (or approximate) our conditions better. More specifically, we highlighted (section 6) that the main difficulty with modern BNN robustness is not coverage (as speculated by some), but rather that approximate inference doesn’t increase the uncertainty fast enough with practical BNN tools (Fig 7). We believe that our observations set major milestones for the community to tackle, and are not of "no real world use".
>
> --
> Minor comments:
> We appreciate the reviewer’s comments that the appendices were clear and helpful for explaining the main ideas, and acknowledge that our notation might be a bit cumbersome at times. We will improve notation as suggested.
>
> * The reviewer mentions in the review the fact that the experiments with HMC included in the paper do not address the data invariance property. This is not the aim of the experiments - rather this is to back up the claim that in the idealised case a BNN increases it’s uncertainty far from the data, as we assume in our idealised model (“we show that near-perfect epistemic uncertainty correlates to density under the image manifold“). It is not obvious a priori that this property is actually true of neural networks in the same way that it is the case for kernel machines with stationary covariance kernels, for example. Previous work on Bayesian techniques for adversarial example detection has invariably used approximate inference, and these proposed defences have generally increased robustness but not eliminated adversarial examples, so we felt that experimentally demonstrating BNNs have this property in the case of idealised *inference* was important.
>
> * “Indeed a model can prevent adversarial examples by predicting high uncertainty for all points that are not near the training examples”: The reviewer correctly points out that a BNN of this kind could fall back to predicting high uncertainty far away from all the data if the invariance property does not hold (e.g a model that predicts P = ½ everywhere is ‘robust’ to adversarial examples but not very interestingly). The reviewer's point that this is a very strong assumption is valid, and is explicitly discussed in the paper (appendix B PROOF CRITIQUE).
>
> * I(w,p) is defined in as equation 1 - we will make this clearer in the text.
>
> * We are uncertain what the reviewer means by VI - as argued by (Gal & Gharimani, 2015) dropout is a form of variational approximation. Other flavours of VI such as mean field exist as well, but based on previous results these are unlikely to be significantly better than dropout unless a far more expressive varational distribution is used, leading to scaling issues (Gal, 2016). We agree that a more scalable method would be highly desirable (as discussed in section 6).

---

> > ### Author Response · Authors · 2018-11-08
> > **Question to reviewer**
> >
> > "Overall, this is a promising theoretical paper although I'm not currently convinced of the real-world applications beyond the somewhat small examples in the experiments section."
> > - Given our answers above, could we ask the reviewer to explain if they think this might still be the case, and if so why?

---

> > > ### Comment · AnonReviewer1 · 2018-11-18
> > > **response to reviewers**
> > >
> > > I want to thank the authors for responding to each of my points, and I wanted to respond to their clarifications.
> > >
> > > I agree with the authors that their paper is the first to provide theory for findings from theoretical papers, namely offering an explanation for why BNNs work well in practice (beyond the non-explanation that "they're Bayesian") and the link between epistemic uncertainty and adversarial examples. As I mentioned in my review, they are indeed addressing important questions posted by previous research in novel ways.
> > >
> > > Appendix C gives an argument for why low test error implies high data coverage (and thus why the data invariance assumption may not be necessary). In Appendix B, the authors argue that the spheres data set, which appears to be a counterexample, may not be representative since in practice we tend to capture this coverage (notably on vision tasks). I believe this claim, but the transformation invariance assumption still exists for non-image domains.
> > >
> > > Another point I wanted to bring up (also brought up by other reviewers) is that real-life adversarial perturbations may be quite small but still larger than the distance for which Bayesian neural networks will assign low value probability. Is there a reason to believe that in practice this distance is covered?
> > >
> > > After reading the other reviews and rebuttals, I see that perhaps the main contribution of the paper is not meant to be technical but rather to aid with future research. That being said, I am still not convinced by the applicability. How much do the authors' contributions actually explain what is seen in practice? If the data invariance assumption can be discounted for applications to images, doesn't the proof boil down to continuity of BNNs (for which delta can be very small)? I stand by my claim that the paper is a promising theoretical paper, although I'm not convinced of the real-world applications of their theory.

---

> > > > ### Author Response · Authors · 2018-11-24
> > > > **response**
> > > >
> > > > We thank the reviewer for taking part in an active discussion with us. We will attempt to answer the reviewers new questions.
> > > >
> > > > “Another point I wanted to bring up (also brought up by other reviewers) is that real-life adversarial perturbations may be quite small but still larger than the distance for which Bayesian neural networks will assign low value probability. Is there a reason to believe that in practice this distance is covered?”
> > > >
> > > > * We are unclear exactly what the reviewer meant in this question - if the reviewer meant the BNN assigning a low *confidence* (rather than low probability, which can still be of high confidence for a certain class), then we actually demonstrate this empirically both in MNIST experiments with HMC (near idealised inference) as well as real-world imagenet-like cats vs dogs classification (last appendix). In both we see that the BNN forces the perturbations to be much larger than otherwise (deterministic model), with the uncertainty being statistically significant to identify often when an image has been manipulated. From the theoretical “guarantee” perspective, this is one of our points we highlight as of interest for future research in S6 - where we exactly identify this as a question the community should concentrate on rather than the question of coverage.

---

### Official Review · AnonReviewer2 · 2018-11-11
**An interesting potential direction the significance of which is still to be demonstrated.**

**Rating:** 5
**Confidence:** 4

**Review:**

The paper studies the adversarial robustness of Bayesian classifiers. The authors state two conditions that they show are provably sufficient for "idealised models" on "idealised datasets" to not have adversarial examples. (In the context of this paper, adversarial examples can either be nearby points that are classified differently with high confidence or points that are "far" from training data and classified with high confidence.) They complement their results with experiments.

I believe that studying Bayesian models and their adversarial robustness is an interesting and promising direction. However I find the current paper lacking both in terms of conceptual and technical contributions.

They consider "idealized Bayesian Neural Networks (BNNs)" to be continuous models with a confidence of 1.0 on the training set. Since these models are continuous, there exists an L2 ball of radius delta_x around each point x, where the classifier has high confidence (say 0.9). This in turn defines a region D' (the training examples plus the L2 balls around them) where the classifier has high confidence. By assuming that an "idealized BNN" has low certainty on all points outside D' they argue that these idealized models do not have adversarial examples. In my opinion, this statement follows directly from definitions and assumptions, hence having little technical depth or value. From a conceptual point of view, I don't see how this argument "explains" anything. It is fairly clear that classifiers only predicting confidently on points _very_ close to training examples will not have high-confidence adversarial examples. How do these results guide our design of ML models? How do they help us understand the shortcomings of our current models?

Moreover, this argument is not directly connected to the accuracy of the model. The idealized models described are essentially only confident in regions very close to the training examples and are thus unlikely to confidently generalize to new, unseen inputs. In order to escape this issue, the authors propose an additional assumption. Namely that idealized models are invariant to a set of transformations T that we expect the model to be also invariant to. Hence by assuming that the "idealized" training set contains at least one input from each "equivalence class", the model will have good "coverage". As far as I understand, this assumption is not connected to the main theorem at all and is mostly a hand-wavy argument. Additionally, I don't see how this assumption is justified. Formally describing the set of invariances we expect natural data to have or even building models that are perfectly encoding these invariances by design is a very challenging problem that is unlikely to have a definite solution. Also, is it natural to expect that for each test input we will have a training input that is close to L2 norm to some transformation of the test input?

Another major issue is that the value of delta_x (the L2 distance around training point x  where the model assigns high confidence) is never discussed. This value is _very_ small for standard NN classifiers (this is what causes adversarial examples in the first place!). How do we expect models to deal with this issue?

The experimental results of the paper are essentially for a toy setting. The dataset considered ("ManifoldMNIST") is essentially synthetic with access to the ground-truth probability of each sample. Moreover, the results on real datasets are unreliable. When evaluating the robustness of a model utilizing dropout, using a single gradient estimation query is not enough. Since the model is randomized, it is necessary to estimate the gradient using multiple queries. By using first-order attacks on these more reliable gradient estimates, an adversary can completely bypass a dropout "defense" (https://arxiv.org/abs/1802.00420).

Overall, I find the contributions of the paper limited both technically and conceptually. I thus recommend rejection.

[UPDATE]: Given the discussion with the authors, I agree that the paper outlines a potentially interesting research direction. As such, I have increased my score from 3 to 5 (and updated the review title). I still do not find the contribution of the paper significant enough to cross the ICLR bar.

Comments to the authors:
-- You cite certain detention methods for adversarial examples (Grosse et al. (2017), Feinman et al. (2017)) that have been shown to be ineffective (that is they can be bypassed by an adaptive attacker) by Carlini and Wagner (https://arxiv.org/abs/1705.07263)
-- The organization of the paper could be improved. I didn't understand what the main contribution was until reading Theorem 1 (this is 6 pages into the paper). The introduction is fairly vague about your contributions. You should consider moving related work later in the paper (most of the discussion there is not directly related to your approach) and potentially shortening your background section.
-- How is the discussion about Gaussian Processes connected with your results?
-- Consider making the two conditions more prominent in the text.
-- In Definition 5, the wording is confusing "We define an idealized BNN to be a Bayesian idealized NN..."

---

> ### Author Response · Authors · 2018-11-16
> **Response to reviewer**
>
> We thank the reviewer for the time spent reading our paper.
>
> We would like to dispute the reviewer’s extremely strong claim of “tautological results of questionable significance”. Our proof is by no means tautological (see below), and a proof does not need to be intricate or over-complicated to shed light on a matter of interest. On the contrary - it is often the simplest of proofs that gives the most interesting insights. We would further like to draw the reviewer’s attention to the fact that this boils down to a matter of presentation. Papers can either over-complicate things to make the results look like an impressive theoretical contribution, or do their best to explain things with the clearest possible presentation. We chose the latter, making an effort to move as much construction outside of the proof into our well-designed premise setup in order to make the the proof clear.
>
> We also strongly disagree with the reviewer’s post-hoc view claiming "this is obvious", a claim which was made with hindsight after reading our proof and problem definition, and which could be made with any proof after understanding its premise. We give many examples of new and previously undiscussed insights (with many highlighted below). These might seem obvious after reading our submission, but these observations are overlooked in the literature, and they have important implications for the way we think about the problem.
> *** We would ask the reviewer explicitly where in the literature they saw the result presented in this work, or any ideas following from the results we presented here?
>
> --
> The rest of the reviewer's criticism falls into two parts - a criticism of the theoretical content of the paper, and of the relevance of the experiments to real world adversarial example defence. We will address these in order.
>
> * "Moreover, this argument [the reviewer's own argument] completely ignores the accuracy of the model. [..] In order to escape this issue, the authors propose [..]"
> - The reviewer brings up an argument and then criticizes their own argument as lacking! (reviewer’s argument: "continuous models with a confidence of 1 on the training set [..] exists an L2 ball [..] the classifier has high confidence [..] low certainty on all points outside D’"). This is exactly the reason why we introduce the concept of T, and which allows us to ***expose a sufficient set of requirements for a model to be robust to adversarial examples***. The reviewer then criticizes the proof as tautological, and says they do not see how the results give anything of value. We mention above that we see Bayesian methods and their robustness as an important direction for future research on adversarial examples, a stance the reviewer agrees with. The purpose of the proof is to illustrate which properties of Bayesian classifiers are necessary for adversarial robustness. Simply being a Bayesian model, for example, is clearly insufficient - linear classifiers will have adversarial examples in high dimensions regardless of whether they are Bayesian or not because the model class is not expressive enough for the uncertainty to increase far away from the data. As mentioned in our reply to the second reviewer, we do not claim that our assumptions are realistic or that our proof is particularly difficult if we assume them.
> However, the assumptions do ***shed light on which properties of Bayesian models are important for being robust to adversarial examples***.
>
> * "this assumption [existence of a set of transformations T both model and data are invariant to] is not connected to the main theorem at all"
> - We are not sure why the reviewer claims that the assumption is not connected to the main theorem? As the reviewer themselves points out, without the assumption of invariance to a class of transformations it is possible for a model which is “uncertain about all inputs not extremely close to a training input” to satisfy our definition of being robust. This is clearly not a very interesting class of models. We introduce the invariance class as a way to avoid this degeneracy. Further, it is, as the reviewer points out, not a very realistic assumption for real models. We do not claim that it is an assumption that holds in practice - we observe that with this assumption we can prove robustness, and from there we proceed to examine the counterpart properties of non-idealised networks (which we discuss in appendix C).
>
> [response continued in a separate message]

---

> > ### Author Response · Authors · 2018-11-16
> > **[continued response]**
> >
> > * "How do these results guide our design of ML models?" / "the value of delta_x is very small"
> > - Both these questions are closely related to each other, and addressed explicitly in our discussion of future research in section 6. We acknowledge that there's very much room for extending our results. However, it is impossible to follow up on all possible threads of future research in a single paper: We must first lay down the foundations for such future research before we can start to pursue it (and we give initial results in our experiments sections). More specifically, given our insights brought in our results, we direct future research to study how different model architectures can yield different uncertainty properties that can increase fast enough / be large for regions far away from the data.
> > In fact, ***one of our main insights/conclusions is that the community's concentration on robustness of models which can generalise beyond “memorising their data” (ie increasing coverage) is misguided, and research should concentrate on model architectures that increase _uncertainty_ fast enough*** (see eg fig 7).
> >
> > * We are slightly confused by some of the reviewers points - they write that the region around each point where the model assigns high confidence is very small for real NNs, which they say causes adversarial examples. But surely the opposite is true - real NNs are very confident about *large* regions of input space which are not close to any training data (see figures 7 and 9 for example). What we assume the reviewer means is that this region could be very small in our idealisation of real NNs. Indeed we observe this to an extent - one notable result from the experiments on manifold mnist (table 1) is that with idealised inference, HMC networks are less robust to *random* noise - that is, they become uncertain quickly when small amounts of noise are added to the input, which deterministic networks do not. These inputs do not follow the data distribution though, and indeed validation error with respect to noise distribution is low for the model.
> >
> > * "The dataset considered ("ManifoldMNIST") is essentially synthetic with access to the ground-truth probability of each sample"
> > - We don't understand how the reviewer sees this as a critical point? It is completely valid to say that these experiments are in a contrived setting, but we do not think this makes them irrelevant. The aim of these experiments is to see whether we can approximate the conditions in our proof, by using MCMC inference, as close as we can get to for an idealised situation with NNs as far as we know. Since this experiment uses HMC both for inference and for evaluating the probability of data under the generative model, it would be difficult to scale to a more realistic synthetic dataset. The relevance of these experiments is that firstly, they demonstrate that (in this dataset at least) adversarial examples are in regions of lower likelihood under the data distribution. This has often been taken as a given in the literature justifying Bayesian methods for adversarial example defence, but there are also counter-examples making other assumptions (Gilmer et. al., the spheres dataset referenced in the paper). Secondly, our experiments demonstrate that for a model with idealised inference, the mutual information is a proxy for the support of the data distribution (alternatively, the input density).
> > This is another example of a novel insight partly derived from our proof (and presented for the first time in the literature as far as we know), and we would argue that showing that this indeed holds in a real experiment is of significant interest. This experiment is indeed still in a toy setting, but it is considerably closer to conditions in real networks than in the proof.
> >
> > * "the results on real datasets are unreliable [..] using a single gradient estimation query is not enough"
> > - The reviewer raises concerns about the difficulties of testing adversarial examples on stochastic networks, as this can lead to gradient masking, making adversarial examples more difficult to find rather than preventing them from existing. We are aware of these issues, and do not claim that the results in this appendix constitute a full defence against adversarial examples on this dataset. We do not think that our result that dropout requires higher perturbation than normal models (though small perturbation examples can still be found) is particularly controversial, and it is in agreement with previous critical papers on the subject  (for example [Carlini and Wagner, 2017]). We are only aiming to show here that ensembling dropout models provides more robustness than a single one, backing up another insight from the visualisations on MNIST in a more realistic situation.
> > Lastly, we would like to stress that we report results on a powerful attack, the momentum method, that won the NIPS attack competition and was designed to be successful against ensembles of models.

---

> > > ### Comment · AnonReviewer2 · 2018-11-17
> > > **Response**
> > >
> > > I thank the authors for responding to my concerns. Given their clarifications, I understand that this paper is not meant as a technical contribution but rather as a conceptual direction for future research. As such, the paper identifies concrete goals to pursue and conducts preliminary experiments. From this point of view, I do agree that the paper makes a meaningful contribution and that my initial score was too harsh. (I want to note however that the paper reads differently than the points made in the author's response and I would encourage the authors to consider changing the narrative at a few places.)
> > >
> > > Nevertheless, I cannot judge the value of the paper based solely on the potential future research that will utilize the proposed framework. I believe it is the responsibility of the authors to clearly demonstrate the potential and applicability of their approach. Thus I still don't consider the paper suitable for ICLR in its current state. I updated my score from 3 to 5 and edited my initial review to reflect this discussion.
> > >
> > > I respond to specific comments below:
> > >
> > > - Let me clarify what I mean by "tautological" and "obvious". I apologize if this came out as too harsh. From my point of view, the main theorem says: "If the classifier has confidence 1 on training examples and low confidence away from training examples, then there are no high-confidence wrong classifications.". Is there some deeper insight that I am missing? If not, then I would argue that the technical depth of the theorem is limited. I want to note that I don't view technical depth as a fundamental requirement for a theorem. My original perception was that you considered the theorem as more of a technical then a perceptual contribution. I never implied that the results were not novel, as they are novel to the best of my knowledge.
> > >
> > > - "this argument completely ignores the accuracy of the model" I apologize for the poor wording, this was not meant as criticism. This sentence is meant to explain why the set T is necessary. Perhaps a better wording would be "the theorem is not directly connected to the accuracy of the model".
> > >
> > > - I thank the authors for clarifying their point. So, if I understand correctly, the concrete practical recommendation is to focus on models that only exhibit high confidence on the training set and then work towards models that incorporate enough invariances of the data distribution to ensure that these models generalize. If this is the case, then I do think that this is an interesting direction to pursue.
> > >
> > > - Let me clarify my point about high confidence regions. Consider a test example that is correctly classified with confidence 1. Consider the radius of the largest L2 ball such that every point in this ball is classified correctly with high confidence. Then that radius is very small. In other words it is very easy to find a nearby misclassification for any point. By the continuity of the classifier there will be low confidence points between the original and misclassified points. Hence the value of delta is typically very small.
> > >
> > > - I agree that the results on the synthetic dataset are encouraging. However, it is hard to draw conclusions about the behavior of real world datasets given these results.
> > >
> > > - Attacks cannot be considered powerful in isolation but only with respect to the particular model they evaluate. Drawing conclusions from evaluations that are not fully reliable can be misleading. This is why I think it is important to evaluate on attacks that accurately reveal the robustness of the model.

---

> > > > ### Author Response · Authors · 2018-11-24
> > > > **response**
> > > >
> > > > We thank the reviewer for taking part in an active discussion with us and for modifying their initial score.
> > > >
> > > > We do see the theorem as, in the reviewers words, more of a perceptual rather than technical contribution (as our title states, **the contribution of the paper is placing forward sufficient conditions** for idealised models to have no adversarial examples). The reviewer raises some valid criticism about whether this was communicated clearly in the main body of the paper, and we would appreciate the reviewer’s feedback on where they think they got confused with the contribution of the paper being otherwise. We would strive to update this before the end of the revision period.
> > > >
> > > >
> > > > In response to some of the specific questions in the last reply from the reviewer:
> > > >
> > > > * ‘The theorem is not directly connected to the accuracy of the model’ - this depends on the definition of adversarial examples that we use. We would argue that adversarial examples are an distinct issue to model accuracy - what would be an ‘adversarial example’ for an input the model already classifies incorrectly? Since adversarial examples are only really suprising for models with high expected accuracy (otherwise normal data is frequently misclassified) we freely assume that the model is perfectly accurate, at least on data sampled from the true distr. The purpose of the transformation set is to allow meaningful generalisation without simply requiring infinite amounts of data. If you allow infinite data, for example, then we could simply use nearest neighbours as our idealised classifier. The introduction of a transformation set is an attempt to introduce a class of models whose generalisation is not totally trivial, though as many of the reviewers have pointed out the condtition is still very strong.
> > > >
> > > >
> > > > * Indeed, for a neural network the value of delta can be very small. As mentioned above, in our experiments with HMC inference on NNs, we see evidence of this being the case.
> > > >
> > > > * It is always fair to be sceptical of results on synthetic or partially synthetic data, but they also allow us to investigate questions we are otherwise unable to - in particular it is impossible to objectively evaluate the likelihood of a datapoint in a real dataset (without making equally unrealistic assumptions).
> > > >
> > > > * Lastly, while it is true that proposed adversarial defences are often not sufficiently carefully evaluated, we would defend our evaluation here: we do not claim to defeat the attack, just that our procedure increases the size of perturbation needed to fool the model, which I believe is in agreement with C&Ws paper. In that paper’s conclusion, the authors say that the dropout based ‘defense’, while bypassable in isolation, does increase the required distortion, which is what we mean by increased robustness, and is what we oberved in our experiments. We will clarify this in our paper.

---

### Author Response · Authors · 2018-12-22
**Summary of reviewers' points and rebuttal**


We summarise the reviewers' remaining points after our rebuttals for the convenience of new readers (together with some more comments):

***

R1 acknowledges after discussing with us:
* "[The] paper is the first to provide theory for findings from theoretical papers, namely offering an explanation for why BNNs work well in practice (beyond the non-explanation that "they're Bayesian") and the link between epistemic uncertainty and adversarial examples. As I mentioned in my review, they are indeed addressing important questions posted by previous research in novel ways."
* "After reading the other reviews and rebuttals, I see that perhaps the main contribution of the paper is not meant to be technical but rather to aid with future research."
* The reviewer still posits "That being said, I am still not convinced by the applicability."
--- We summarise the main points on applicability we presented to the reviewer:
--- First, for the first time in the literature (as far as we know), we studied in detail a proposed mechanism to explain why BNNs have been observed to be empirically more robust to adversarial examples.
--- Second, for the first time in the literature, we proved (and gave empirical evidence in Fig 3) that a connection exists between epistemic uncertainty/input density and adversarial examples, a link which was only *hypothesised* previously.
--- Further, our proof highlighted sufficient conditions for robustness which resolved the inconsistency between (Nguyen et al., 2015; Li, 2018) and (Gilmer et al., 2018).
--- Apart from resolving an open dispute in the literature, our observations above also suggest future research directions towards tools which answer our conditions better. We believe that our observations set major milestones for the community to tackle.
--- Lastly, we proposed new synthetic datasets for the community to experiment with adversarial examples (where we can calculate ground truth image density, something which was only done in hand-wavey ways so far by looking at images by hand). We also demonstrated our ideas in ***real world imagenet-like tasks***, and in the process also exposed new attacks and defences on Bayesian techniques - see section 5.

***

R2 acknowledges after discussing with us:
* "[Results] are novel to the best of my knowledge"
* "I do agree that the paper makes a meaningful contribution" / "the paper identifies concrete goals to pursue", with the remaining criticism that "I cannot judge the value of the paper based solely on the potential future research that will utilize the proposed framework".
--- We would highlight that we give many examples of new and previously undiscussed insights (with many also highlighted through the rebuttal). These observations are overlooked in the literature, and they have important implications for the way we think about the problem.

***

R3 acknowledges after discussing with us:
* "I didn’t read the appendix for my original review. Appendix C has several discussions about these conditions [which the reviewer got confused about], which in my opinion are interesting and important, and should be in the main content"
* "I would like to emphasize that the presentation of this paper needs to be improved." Eg "definition of the \delta ball" and others
--- we did not think some standard terms merited their own definition. Instead we put this in a footnote to clarify for non-technical readers.

---

### Meta-Review · Area_Chair1 · 2018-12-16
**An intriguing idea but the exact impact is unclear**

**Confidence:** 4
**Recommendation:** Reject

**Metareview:**

This paper conducts a study of the adversarial robustness of Bayesian Neural Network models. The reviewers all agree that the paper presents an interesting direction, with sound theoretical backing. However, there are important concerns regarding the significance and clarity of the work. In particular, the paper would greatly benefit from more demonstrated empirical significance, and more polished definitions and theoretical results.